# Anti-Staphylococcal, Anti-Candida, and Free-Radical Scavenging Potential of Soil Fungal Metabolites: A Study Supported by Phenolic Characterization and Molecular Docking Analysis

**Amal A. Al Mousa** [1,*] **, Mohamed E. Abouelela** [2] **, Nadaa S. Al Ghamidi** [1] **, Youssef Abo-Dahab** [3] **, Hassan Mohamed** [4] **, Nageh F. Abo-Dahab** [4] **and Abdallah M. A. Hassane** [4,*]

[1] Department of Botany and Microbiology, College of Science, King Saud University, P.O. Box 145111, Riyadh 4545, Saudi Arabia; nalgamdi1@ksu.edu.sa

[2] Department of Pharmacognosy, Faculty of Pharmacy (Boys), Al-Azhar University, Cairo P.O. Box 11884, Egypt; m_abouelela@azhar.edu.eg

[3] Ministry of Health and Populations, Cairo P.O. Box 11516, Egypt; youssef.nageh@live.com

[4] Botany and Microbiology Department, Faculty of Science, Al-Azhar University, Assiut 71524, Egypt; hassanmohamed85@azhar.edu.eg (H.M.); abodahabn@azhar.edu.eg (N.F.A.-D.)

* Correspondence: aalmosa@ksu.edu.sa (A.A.A.M.); abdallahhassane@azhar.edu.eg (A.M.A.H.); Tel.: +966-505389419 (A.A.A.M.); +20-1122990125 (A.M.A.H.)

**Abstract:** *Staphylococcus* and *Candida* are recognized as causative agents in numerous diseases, and the rise of multidrug-resistant strains emphasizes the need to explore natural sources, such as fungi, for effective antimicrobial agents. This study aims to assess the in vitro anti-staphylococcal and anti-candidal potential of ethyl acetate extracts from various soil-derived fungal isolates. The investigation includes isolating and identifying fungal strains as well as determining their antioxidative activities, characterizing their phenolic substances through HPLC analysis, and conducting in silico molecular docking assessments of the phenolics' binding affinities to the target proteins, *Staphylococcus aureus* tyrosyl-tRNA synthetase and *Candida albicans* secreted aspartic protease 2. Out of nine fungal species tested, two highly potent isolates were identified through ITS ribosomal gene sequencing: *Aspergillus terreus* AUMC 15447 and *A. nidulans* AUMC 15444. Results indicated that *A. terreus* AUMC 15447 and *A. nidulans* AUMC 15444 extracts effectively inhibited *S. aureus* (concentration range: 25–0.39 mg/mL), with the *A. nidulans* AUMC 15444 extract demonstrating significant suppression of *Candida* spp. (concentration range: 3.125–0.39 mg/mL). The *A. terreus* AUMC 15447 extract exhibited an $IC_{50}$ of 0.47 mg/mL toward DPPH radical-scavenging activity. HPLC analysis of the fungal extracts, employing 18 standards, revealed varying degrees of detected phenolics in terms of their presence and quantities. Docking investigations highlighted rutin as a potent inhibitor, showing high affinity (−16.43 kcal/mol and −12.35 kcal/mol) for *S. aureus* tyrosyl-tRNA synthetase and *C. albicans* secreted aspartic protease 2, respectively. The findings suggest that fungal metabolites, particularly phenolics, hold significant promise for the development of safe medications to combat pathogenic infections.

**Keywords:** fungi; phenolics; anti-staphylococcus; anti-candidal; molecular docking

## 1. Introduction

Soil contains a diverse range of microorganisms, including fungi, which are the most abundant producers of potentially beneficial biologically active metabolites [1]. Out of all the bioactive compounds that have been isolated from biological sources, more than 38% of the metabolites that are biologically active come from fungi [2]. It has been reported that fungi are potent producer of diverse bioactive metabolites, with their secondary metabolism comprising polysaccharides [3], enzymes [4,5], lipids and fatty acids [6], and low molecular-weight byproducts that are produced as an adaptation for specific functions in nature [7]. These metabolites may also include alkaloids, flavonoids, benzopyranones,

phenolic acids, tannins, quinones, saponins, terpenoids, steroids, tetralones, xanthones, and many others [8]. The identified metabolites showed varied bioactivities with anticancer [9], antioxidative [10,11], antifungal [12,13], immunosuppressive [14], antiprotozoal [15], antiviral [16], and antibacterial efficiencies [17]. However, harmful influences can be caused by mycotoxins produced by some types of fungi [18,19]. Due to their unique metabolism, fungi are able to produce a wide range of useful secondary metabolites, which increases their potential for drug development [20].

Phenolics are compounds that include one or more phenol units and are primarily found in plants, although they can also be found in bacteria and fungi [21]. For example, tyrosol and p-hydroxyphenylacetamide isolated from the extract of the endophytic fungus *Coriolopsis rigida* showed potent antioxidative activity [22]. The significance of phenolic compounds like vanillic and caffeic acids in potentially overcoming resistance in multidrug-resistant bacteria like methicillin-resistant *Staphylococcus aureus* (MRSA) has been highlighted in a numerous research studies [23,24] elucidating the hopeful impact of phenolic acids in suppressing the emergence of new resistant strains [25]. Flavonoids and isoflavonoids are other promising antimicrobial mediators that target many microbial cells and reduce the virulence particularity of drug-resistant strains [26].

The ability of *Candida* species to cause infections ranging from superficial to potentially life-threatening is well documented [27,28]. This variability in infection severity reflects the complexity and adaptability of these pathogenic organisms. *Candida albicans* is the most widespread species, while the ratio of non-albicans species has grown over the past few decades, comprising *C. glabrata*, *C. krusei*, *C. parapsilosis*, and *C. tropicalis* [29–31]. *Candida albicans* is a polymorphic and pathogenic fungal species causing enormous damage to humans, including biofilm formation, and oral, vaginal, and skin infections in immune-deficient patients [32,33]. *Candida albicans* can grow together with *S. aureus* [34] and other diverse bacteria or fungi in oral as well as in skin/wound infections [35]. *Staphylococcus aureus* serves as a commensal bacterium in a significant portion (approximately 30%) of the human population [36]. It commonly colonizes the skin and is associated with infections in wounds and deep tissues [37]. The bacterium has demonstrated remarkable genetic adaptability, leading to the emergence of various drug-resistant strains that display resistance to existing antimicrobial medications. This resistance poses a challenge to current treatment options [38].

*Staphylococcus aureus* is frequently found alongside *C. albicans* in systemic bloodstream infections, with over 25% of candidemia cases estimated to be polymicrobial, whereas systemic *Staphylococcus* infections without *Candida* are less prevalent [39,40]. The emergence of azole-resistant *C. albicans* isolates has prompted various suggested approaches to address this issue. Similarly, effectively treating drug-resistant *S. aureus* poses a significant challenge in the field of medicine [41]. Growing fungal resistance limits their convenient therapeutic efficacies, thus making the treatment of fungal infection disease more intractable [42]. MRSA strains are intrinsically resistant to β-lactams and rapidly develop resistance to multiple antimicrobial drug classes [43]. The widespread use of antibiotics has led to the prominent problem of microbial pathogens developing drug resistance, underscoring the urgent need to explore new antibacterial and antifungal agents that are both safe for patients and capable of efficiently eliminating pathogens to promote wound healing [44].

The genetic plasticity of *C. albicans* is directly reflected in its metabolic diversity and its ability to produce a wide range of virulence factors, such as secreted aspartyl proteases (SAPs), which cleave the extracellular matrix and fluid-phase proteins to induce tissue damage and promote infection [33,45,46]. Secreted aspartic proteinase 2 (SAP2) represents an important virulence factor for vaginal infection, which is released by the yeast cells and causes damage to the reconstituted human epithelium [47,48]. As a result, inhibiting the enzyme's active center using phytochemicals would reduce the severity of the enzyme's virulence [32]. *Staphylococcus aureus* tyrosyl-tRNA synthetase belongs to the group of aminoacyl-tRNA synthetases, which are responsible for catalyzing the covalent binding of amino acids to their corresponding tRNA by generating charged tRNAs and

which play an important role in the production of proteins [49]. The structural differences between bacterial and eukaryotic aaRSs are a unique feature that allows the consideration of tyrosyl-tRNA synthetase as a good therapeutic target for the prevention of bacterial infection [50,51]. Thus, inhibition of this enzyme influences cell proliferation due to its vital role in the biosynthesis of proteins [52].

Progress in analytical chemistry, computational tools, and drug discovery research has facilitated the creation of fungal-derived antimicrobial compounds. These compounds exhibit potential therapeutic effects and can be utilized independently or in combination therapies to manage challenging pathogens that are resistant to traditional treatments [53]. Moreover, structure-based drug design plays a pivotal role [54]. Molecular docking is performed to check the reliability of compounds in protein-binding sites [55]. The present study focused on the in vitro antimicrobial potency of soil fungal extracts, which are rich in phenolic constituents, from different *S. aureus* and *Candida* species isolates. HPLC analysis of phenolic compounds as well as their in silico analysis in comparison with *S. aureus* tyrosyl-tRNA synthetase and *C. albicans* secreted aspartic protease 2 will be conducted to evaluate the interactions of these bioactive compounds and to provide clues for the therapeutic targeting of the enzymes' active sites.

## 2. Materials and Methods

### 2.1. Sample Collection and Fungal Isolation

Seven soil samples were randomly collected from Ha'il, (27.8942° N, 42.6832° E), Saudi Arabia, during December 2022, from the superficial layer of the soil with a maximum depth of 10 cm and transported in sterile plastic bags to the laboratory. Mycological analysis was achieved by the dilution-plate method [56] and Czapek's agar medium was utilized for the isolation and purification of fungal isolates.

### 2.2. Identification of Fungal Isolates

The most bioactive fungal isolates were identified based on their macromorphology and microscopic features using the key references of Raper and Fennell [57], Moubasher [58] and Domsch et al. [59]; these were then deposited into the Assiut University Mycological Center (AUMC) culture collection with an institutional number.

### 2.3. Fermentation and Extraction of Fungal Metabolites

The fermentation process for the fungal isolates was performed in autoclaved flasks containing solid rice inoculated with 1 mL of the fungal spore suspension ($10^7$ spore/mL), and then incubated at $28 \pm 1$ °C for thirty days [60]. The fermented medium was extracted twice by ethyl acetate (EtOAc) (Analytical Grade, Alpha Chemika, Mumbai, India) [61], filtered over $Na_2SO_4$ anhydrous (AL-Nasr Chemicals Co., Cairo, Egypt), and concentrated by a vacu-rotavapor [62].

### 2.4. Antimicrobial Susceptibility Testing

2.4.1. Test Microorganisms

Test microbial strains and isolates were obtained from the culture collection at the Botany and Microbiology Department, Faculty of Science, Assiut, Al-Azhar University, Egypt. The tested *Staphylococcus aureus* strains included *S. aureus* ATCC 6538, enterotoxigenic *S. aureus* AZHAR2 (MF563554) [63], *S. aureus* AUHL123 [64], *S. aureus* BLBM 112, and *S. aureus* MLBM 117, which were grown on nutrient agar medium (NA) at 37 °C for 24 h. The *Candida* spp. comprised *C. albicans* ATCC 10231, *C. albicans* MLBM 73 (ON430508) from urine samples, *C. albicans* C-13 (MN419371), *C. glabrata* C-4 (MN419362), and *C. krusei* C-30 (MN419388) from vaginal swab specimens [65], and these were grown on Sabouraud dextrose agar medium (SDA) for 2 days at 25 °C.

### 2.4.2. Agar Well-Diffusion Method

The preliminary anti-staphylococcal and anti-candidal activities of fungal EtOAc crude extracts were screened by the well-diffusion method [66] on NA and SDA, respectively. One hundred mg of each fungal extract was dissolved in 1 mL of dimethyl sulfoxide (DMSO) (MilliporeSigma—St. Louis, MI, USA). Wells of 8 mm diameter were made in the media plates, pre-inoculated with the staphylococcal and candidal spore suspension ($10^5$ CFU/mL), filled with 100 µL of the extract (10% *w/v*), and then incubated for 24 h at 37 °C (bacteria) or and at 28 °C for 2 days (*Candida*). The diameter of the inhibition zone around the well was measured in millimeters (mm) using vernier calipers. Chloramphenicol (Chemical Industries Development Company, Cairo, Egypt) (0.1% *w/v*) was utilized in the anti-staphylococcal test as a positive control, while 0.1% *w/v* fluconazole (Pfizer, Egypt) was used for candida. DMSO was used as a negative control, and all experiments were performed three times. Extracts that showed positive activities were subjected to minimum inhibitory concentration (MIC) determination.

### 2.4.3. Determination of Minimum Inhibitory Concentration

A microdilution assay combined with *p*-iodonitrotetrazolium chloride (INT) (Merck, Darmstadt, Germany) at 0.2 mg/mL was used to determine the MIC values of the active EtOAc fungal extracts on the tested *S. aureus*. A 96-well microtiter plate was filled with a final volume of 200 µL in each well, comprising 100 µL of $10^5$ CFU/mL and 100 µL of 2-fold serially diluted fungal extract in nutrient broth, which was incubated at 37 °C for 24 h. The final fungal extract concentration, using two-fold serial dilution, in each well was 100, 50, 25, 12.5, 6.25, 3.125, 1.56, 0.78, 0.39, 0.195, and 0.098 mg/mL, respectively. Positive and negative controls (wells with and without chloramphenicol, 2-fold serially diluted) and blank (non-inoculated wells) were included. After incubation, 40 µL of INT was added to the wells and set aside for 30 min at 37 °C, and then the MICs of the extracts were determined. Lall et al. defined the MIC as the lowest concentration at which INT was reduced to formazan due to mitochondrial dehydrogenase in the bacterial cell, which results in a color change from yellow to purple [67]. The minimum bactericidal concentration (MBC) was assessed by streaking 50 µL from each well of the determined MIC and higher concentrations to NA plates, and these were then examined after incubation for one day at 37 °C.

*Candida* spp. cell suspension was adjusted to $10^5$ cells/mL in sterile Sabouraud dextrose broth, and then 100 µL of this was added to a 96-well microtiter plate containing 100 µL aliquots of 2-fold serially diluted fungal extract in Sabouraud dextrose broth. Positive (wells with fluconazole, 2-fold serially diluted) and negative (wells without any treatment) controls as well as blank (wells with uninoculated medium) were established. The plates were incubated at 30 °C for 48 h, and then the MIC values were recorded visually, based on the growth of the microorganism. The MIC was defined as the lowest concentration at which no visible growth was established [68]. The minimum fungicidal concentration (MFC) was confirmed by transferring 50 µL from the wells of the previously assessed MIC and of higher concentrations to SDA plates and then observed after incubation for two days at 30 °C.

### 2.5. Free-Radical-Scavenging Assay

The radical-scavenging assay was performed by using 1,1-diphenyl-2-picryl-hydrazyl (DPPH) (MilliporeSigma—St. Louis, USA) based on the procedures described in [69]. Several concentrations of fungal extracts ranging between 10 and 0.1 mg/mL were prepared, and 0.2 mL of each extract was added to 1.8 mL of 0.1 mM methanolic DPPH (negative control). These were measured spectrophotometrically after 30 min 517 nm and compared with the blank. To determine the scavenging capacity, the following formula was carried out:

$$\%\text{Antioxidative capacity} = \frac{\text{A0} - A}{\text{A0}} \times 100$$

where A0 represents the absorbance of the negative control and *A* is the absorbance of the tested extract. The $IC_{50}$ was derived by intercalation from the linear regression exploration.

### 2.6. Brine-Shrimp Lethality Assay

Fungal extract cytotoxicity was assessed on *Artemia salina* larvae by the addition of 100 μL from each extract dilution to tubes containing 0.9 mL seawater with 10 living larvae for 1 day. After that, the number of vital larvae in each tube was counted and the average mortality percentage for each extract was calculated along with the $LC_{50}$ (extract concentration which caused 50% larval mortality) [70].

### 2.7. Phylogenetic Analysis of the Most Potent Isolates
2.7.1. Isolation of Genomic DNA

Total DNA was extracted from the pure fungal culture, which had been cultivated in Czapeks' broth for 5 days at 28 °C, using the Norgen Plant/Fungi DNA Isolation Kit (Sigma, Thorold, ON, Canada) and preserved at −20 °C [71].

2.7.2. PCR Amplification and Nucleotide Sequence Analysis

As described by Mohamed et al. [72], the internal transcribed spacer (ITS) region of the tested strain was amplified using the specific universal primers ITS-1 (5′-TCC GTA GGT GAA CCT GCG G-3′) and ITS-4 (5′-TCC TCC GCT TAT TGA TAT GC-3′), while the PCR was performed as described previously by Hassan et al. [71]. The obtained sequences were aligned with the BLAST search tool from NCBI to reveal any similarities. The BioEdit software program (version No. 7.2.5) was employed to check and analyze the ITS sequences, and the search for homology was achieved by comparison with strains from sequencing databases using the BLAST search algorithm of GenBank (http://www.ncbi.nlm.nih.gov/BLAST/ (accessed on 13 May 2023)).

### 2.8. Determination of Total Phenolic and Flavonoid Content

The total phenolic content (TPC) was assessed spectrophotometrically three times in accordance with Suleria et al. [73], with the procedure modified by adding a 0.5 mL volume (1% *w/v*) of the extract to Folin–Ciocalteu's phenol reagent (MilliporeSigma—St. Louis, CA, USA) (0.5 mL) and 1 mL of sodium carbonate (10% *w/v*) and measuring at 750 nm after 1 h against the blank. The TPC was expressed as gallic acid equivalents (mg/g) through the standardization curve equation. Assessment of the total flavonoid content (TFC) was conducted spectrophotometrically as reported by Quettier-Deleu et al. [74] by mixing a 0.5 mL extract (1% *w/v*) with 1 mL of ethanolic aluminum chloride (2% *w/v*) (AL-Nasr Chemicals Co., Egypt) and measured after 10 min at 430 nm against the blank. The TFC was calculated as quercetin equivalents (mg/g) by employing the standardization curve equation.

### 2.9. High-Performance Liquid Chromatography Analysis

The analysis employed an Agilent 1260 HPLC-DAD system (Agilent Technologies, Waldbronn, Germany) for high-performance liquid chromatography (HPLC). A separation process was conducted using an Eclipse C18 column (4.6 mm × 250 mm i.d., 5 μm), with 10 μL of each fungal extract sample injected. The mobile phase comprising water and 0.05% trifluoroacetic acid in acetonitrile (HPLC grade, Alpha Chemika, Mumbai, India), flowed at a rate of 1 mL/min, following a linear gradient. The column temperature was approximately 35 °C, and a multi-wavelength detector performed the monitoring at 280 nm [75]. Cinnamic, caffeic, ferulic, syringic, gallic, ellagic, *p*-coumaric, and chlorogenic acids, in addition to hesperetin, kaempferol, rutin, catechin, quercetin, apigenin, methyl gallate, vanillin, daidzein, and naringenin were purchased from Merck, Darmstadt, Germany, and utilized as standard phenolics.

*2.10. Molecular Docking Simulations with Target Proteins*

The docking experiments were carried out using the PyRx software version 0.9 [76]. The inhibition of *S. aureus* tyrosyl-tRNA synthetase (PDB ID: 1JIJ) [77] and *C. albicans* secreted aspartic protease 2 (PDB ID: 3PVK) [78] by the eighteen detected compounds were assessed by evaluating their ligand–protein binding patterns and interactions with enzymes retrieved from the Protein Data Bank (http://www.rcsb.org/pdb (accessed on 20 June 2023)). The protein targets were prepared for docking by removing unnecessary water molecules. The active site for interactions was selected as the complex inhibitor ligand site. Further, the detected compounds were justified, and a virtual ligand database was generated. The docking scores were recorded by a rigid receptor–flexible ligand-docking procedure and 2D and 3D interaction figures were generated by BIOVIA Discovery Studio (v21.1.0.20298) [76,79].

*2.11. Data Analysis*

All experiments were carried out three times. Data were presented as the mean $\pm$ SD and established by analysis of variance (one-way ANOVA) using the SPSS software, version 16 (IBM, Armonk, NY, USA), as being below the 0.05 level of significance.

## 3. Results

*3.1. Identification of Fungal Isolates*

Forty-five fungal isolates were derived from different collected soil samples; among these isolates, nine fungal ethyl acetate extracts exhibited anti-staphylococcal and anti-*Candida* activities. These potent isolates were identified based on their macro- and microscopic characteristics as *Trichoderma harzianum* AUMC 15440 and AUMC 15443, *Aspergillus aureolatus* AUMC 15441 and AUMC 15446, *A. nidulans* AUMC 15444, *A. terreus* AUMC 15447 and AUMC 15448, *Penicillium crustosum* AUMC 15445, and *P. novae-zeelandiae* AUMC 15442 (Table 1).

**Table 1.** Bioactive fungal species isolated from different soil samples.

| AUMC No. | Fungal Isolate Identification | Fungal Phylum; Class; Order |
|---|---|---|
| 15440 | *Trichoderma harzianum* Rifai | Ascomycota; Sordariomycetes; Hypocreales |
| 15443 | *T. harzianum* Rifai | |
| 15441 | *Aspergillus aureolatus* Munt.-Cvetk. & Bata | |
| 15446 | *A. aureolatus* Munt.-Cvetk. & Bata | |
| 15444 | *A. nidulans* (Eidam) Wint. | Ascomycota; Eurotiomycetes; Eurotiales |
| 15447 | *A. terreus* Thom | |
| 15448 | *A. terreus* Thom | |
| 15445 | *Penicillium crustosum* Thom | |
| 15442 | *P. novae-zeelandiae* Van Beyma | |

*3.2. Antimicrobial Activity*

As depicted in Table 2, the investigation into the anti-staphylococcal effects of diverse fungal extracts at a concentration of 100 mg/mL demonstrated notable efficacy against the tested *S. aureus*. Notably, among the *S. aureus* isolates, including ATCC 6538, AUHL123, and MLBM 117, the ethyl acetate (EtOAc) extracts derived from *A. terreus* AUMC 15447 and *A. nidulans* AUMC 15444 exhibited significant potency. For AUMC 15447, the inhibition zone diameters were 22.33 mm, 24.67 mm, and 23.00 mm, respectively, compared with the control. Similarly, for AUMC 15444, the inhibition zone diameters were 23.33 mm, 19.67 mm, and 20.33 mm, respectively, demonstrating substantial inhibitory activity compared with the control. On the contrary, some extracts exhibited diverse levels of activity, including instances of no inhibitory effects.

**Table 2.** Anti-staphylococcal activity of the fungal extracts at a concentration of 100 mg/mL.

| Fungal Extract AUMC No. | Diameter of Inhibition Zone (mm) | | | | |
|---|---|---|---|---|---|
| | *S. aureus* ATCC 6538 | *S. aureus* AZHAR2 | *S. aureus* BLBM 112 | *S. aureus* AUHL 123 | *S. aureus* MLBM 117 |
| 15440 | 19.67 ± 0.33 [c] | 11.67 ± 0.33 [c] | 13.67 ± 0.33 [e] | 17.33 ± 0.33 [b,c] | 16.33 ± 0.88 [c,d] |
| 15443 | 11.00 ± 0.58 [d] | 14.33 ± 0.67 [b,c] | 17.33 ± 0.67 [b,c,d] | 14.33 ± 0.33 [c] | 14.00 ± 0.58 [d] |
| 15441 | R | R | R | 10.00 ± 0.58 [d] | R |
| 15446 | 10.00 ± 0.00 [d] | 14.00 ± 0.58 [b,c] | 14.67 ± 0.33 [d,e] | 15.00 ± 0.00 [c] | 12.67 ± 0.67 [e] |
| 15444 | 23.33 ± 0.33 [b] | 16.33 ± 0.88 [b] | 18.00 ± 0.58 [b] | 19.67 ± 0.88 [b] | 20.33 ± 0.33 [b] |
| 15447 | 22.33 ± 0.67 [b] | 16.33 ± 0.33 [b] | 17.67 ± 0.67 [b,c] | 24.67 ± 0.33 [a] | 23.00 ± 0.58 [a,b] |
| 15448 | 17.67 ± 0.33 [c] | 14.67 ± 0.33 [b] | 15.00 ± 0.00 [c,d,e] | 19.33 ± 0.88 [b] | 15.33 ± 0.33 [c,d] |
| 15445 | 9.67 ± 0.33 [d] | 13.67 ± 0.88 [b,c] | 12.67 ± 0.33 [e] | 16.67 ± 0.88 [b,c] | 17.00 ± 0.00 [c] |
| 15442 | R | R | R | R | R |
| Chloramphenicol | 26.00 ± 0.58 [a] | 27.33 ± 0.33 [a] | 27.67 ± 1.20 [a] | 26.00 ± 1.53 [a] | 25.33 ± 0.88 [a] |

R, resistant. Data are presented as the mean ± SD, and values associated with superscripts differ significantly, with $p < 0.05$.

Regarding the anti-candidal effects of the fungal extracts, the EtOAc extract from *A. nidulans* AUMC 15444 stood out as the most effective, demonstrating inhibition zone diameters of 20.67 mm, 19.67 mm, 22.33 mm, 20.00 mm, and 24.00 mm against various tested *Candida* species. Additionally, *T. harzianum* AUMC 15440 showed significant activity against both *C. albicans* ATCC 10231 and *C. albicans* MLBM 73 compared with the control (Table 3). In contrast, the remaining extracts displayed varying or no inhibition against the tested *Candida* species. It is noteworthy that certain fungal extracts exhibited antimicrobial efficacy against both *Staphylococcus* and *Candida* species, while others were effective against only one of these pathogens.

**Table 3.** Anti-candidal activity of the fungal extracts at a concentration of 100 mg/mL.

| Fungal Extract AUMC No. | Diameter of Inhibition Zone (mm) | | | | |
|---|---|---|---|---|---|
| | *C. albicans* ATCC 10231 | *C. albicans* MLBM 73 | *C. albicans* C-13 | *C. glabrata* C-4 | *C. krusei* C-30 |
| 15440 | 20.33 ± 0.88 [b] | 19.33 ± 0.33 [b] | 13.67 ± 0.33 [b] | 13.00 ± 0.58 [b] | R |
| 15443 | 9.67 ± 0.33 [d,e] | R | R | R | R |
| 15441 | 9.00 ± 0.00 [e] | 9.33 ± 0.33 [c] | R | 10.00 ± 0.00 [c] | R |
| 15446 | 12.00 ± 0.00 [c,d] | R | R | R | R |
| 15444 | 20.67 ± 0.67 [b] | 19.67 ± 0.33 [b] | 22.33 ± 0.88 [a] | 20.00 ± 0.58 [a] | 24.00 ± 0.58 [a] |
| 15447 | 13.00 ± 0.00 [c] | R | 12.00 ± 0.58 [b] | R | R |
| 15448 | 13.33 ± 0.33 [c] | R | R | R | 11.00 ± 0.58 [b] |
| 15445 | R | R | R | R | R |
| 15442 | 12.33 ± 0.33 [c,d] | 11.00 ± 0.58 [c] | 10.67 ± 0.33 [b] | 10.00 ± 0.58 [c] | R |
| Fluconazole | 24.33 ± 1.20 [a] | 26.00 ± 0.58 [a] | 24.33 ± 0.88 [a] | 21.67 ± 0.88 [a] | 22.67 ± 0.33 [a] |

R, resistant. Data are represented as the mean ± SD, and values associated with superscripts differ significantly, with $p < 0.05$.

Determination of the Minimum Inhibitory Concentration (MIC) of Extracts

The MIC, minimum bactericidal concentration (MBC), and minimum fungicidal concentration (MFC) of the bioactive fungal extracts were assessed utilizing the microdilution assay. For *S. aureus*, the MIC and MBC were determined employing INT, and the results showed a great variation among the values in the range of the tested concentrations (100 to 0.39 mg/mL). Both *A. terreus* AUMC 15447 and *A. nidulans* AUMC 15444 extracts showed the most significant MIC values against *Staphylococcus* isolates in the range between 25 and 0.39 mg/mL, followed by extracts from *T. harzianum* AUMC 15440 and *A. terreus* AUMC 15448 (Table 4). It was noted that the obtained MFC values were represented by the concentrations preceding the MIC. The determined MIC and MFC of extracts against the *Candida* spp. revealed the high potency of the *A.*

*nidulans* AUMC 15444 extracts, where the MIC range was between 3.125 and 0.39 mg/mL, while the MFC was in the range between 6.25 and 0.78 mg/mL (Table 5).

**Table 4.** Anti-staphylococcal activity assay and MIC and MBC (mg/mL) determination of the tested fungal extracts.

| Fungal Extract AUMC No. | MIC and MBC (mg/mL) | | | | | | | | | |
|---|---|---|---|---|---|---|---|---|---|---|
| | ATCC 6538 | | AZHAR2 | | BLBM 112 | | AUHL 123 | | MLBM 117 | |
| | MIC | MBC | MIC | MBC | MIC | MBC | MIC | MBC | MIC | MBC |
| 15440 | 3.125 | 6.25 | 50 | 100 | 25 | 50 | 3.125 | 6.25 | 3.125 | 6.25 |
| 15443 | 100 | R | 12.5 | 25 | 3.125 | 6.25 | 25 | 50 | 25 | 50 |
| 15441 | R | R | R | R | R | R | 100 | R | R | R |
| 15446 | 100 | R | 25 | 50 | 25 | 50 | 25 | 50 | 50 | 100 |
| 15444 | 0.78 | 1.56 | 25 | 50 | 12.5 | 25 | 0.78 | 1.56 | 0.78 | 1.56 |
| 15447 | 0.78 | 1.56 | 12.5 | 25 | 6.25 | 12.5 | 0.39 | 0.78 | 0.39 | 0.78 |
| 15448 | 6.25 | 12.5 | 12.5 | 25 | 12.5 | 25 | 3.125 | 6.25 | 12.5 | 25 |
| 15445 | 100 | R | 25 | 50 | 25 | 50 | 12.5 | 25 | 6.25 | 12.5 |
| 15442 | R | R | R | R | R | R | R | R | R | R |
| Chloramphenicol | 0.39 | 0.78 | 0.78 | 1.56 | 0.78 | 1.56 | 0.195 | 0.39 | 0.195 | 0.39 |

R, resistant.

**Table 5.** Anti-candidal activity assay and MIC and MBC (mg/mL) determination of the tested fungal extracts.

| Fungal Extract AUMC No. | MIC and MBC (mg/mL) | | | | | | | | | |
|---|---|---|---|---|---|---|---|---|---|---|
| | ATCC 10231 | | MLBM 73 | | C-13 | | C-4 | | C-30 | |
| | MIC | MFC | MIC | MFC | MIC | MFC | MIC | MFC | MIC | MFC |
| 15440 | 1.56 | 3.125 | 3.125 | 6.25 | 6.25 | 12.5 | 12.5 | 25 | R | R |
| 15443 | 100 | R | R | R | R | R | R | R | R | R |
| 15441 | 100 | R | 100 | R | R | R | 100 | R | R | R |
| 15446 | 50 | 100 | R | R | R | R | R | R | R | R |
| 15444 | 1.56 | 3.125 | 1.56 | 3.125 | 0.78 | 1.56 | 3.125 | 6.25 | 0.39 | 0.78 |
| 15447 | 25 | 50 | R | R | 12.5 | 25 | R | R | R | R |
| 15448 | 12.5 | 25 | R | R | R | R | R | R | 50 | 100 |
| 15445 | R | R | R | R | R | R | R | R | R | R |
| 15442 | 25 | 50 | 100 | R | 100 | R | 100 | R | R | R |
| Fluconazole | 0.78 | 1.56 | 1.56 | 3.125 | 0.78 | 1.56 | 0.39 | 0.78 | 0.78 | 1.56 |

R, resistant.

### *3.3. Antioxidative and Cytotoxic Activities of Extracts*

The results illustrated that the lowest $IC_{50}$ value (0.47 mg/mL) in the DPPH antioxidant assay was afforded by *A. terreus* AUMC 15447 followed by that of the *A. terreus* AUMC 15448 EtOAc extract (0.58 mg/mL), while the highest $IC_{50}$ value (42.87 mg/mL) was scored by *T. harzianum* AUMC 15443. Regarding the *Artemia* cytotoxicity assay, the best (lowest $LC_{50}$ value) of 1.18 mg/mL was recorded by the *T. harzianum* AUMC 15440 extract followed by the *A. terreus* AUMC 15448 extract (1.29 mg/mL), whereas the highest $LC_{50}$ values (2.01 and 2.0 mg/mL) were presented by extracts from *T. harzianum* AUMC 15443 and *A. terreus* AUMC 15447, respectively (Table 6).

### *3.4. Molecular Identification of the of Most Potent Isolates*

Identification of the most biopotent isolates was performed molecularly by sequencing the ITS region, and then sequences were subjected to a BLAST search of the NCBI database. The isolates were confirmed as *Aspergillus nidulans* AUMC 15444 (GenBank accession no. OR064351) and *Aspergillus terreus* AUMC 15447 (GenBank accession no. OR064355). Nucleotide comparison of the ITS regions among *A. nidulans* AUMC 15444 strain and

other similar strains recaptured from the NCBI showed 99.62–99.81% identity and 99–100% coverage with several strains of the same species. *Penicillium chrysogenum* was included as an outgroup strain (Figure 1a). Meanwhile, *A. terreus* AUMC 15447 showed 99.67–100% identity and 98–100% coverage with numerous strains of the same species comprising the type strain *A. terreus* ATCC1012 with GenBank accession no. NR_131276. *Aspergillus ochraceus* was included as an outgroup strain (Figure 1b).

**Table 6.** Antioxidative activities ($IC_{50}$ values) by the DPPH assay and the brine-shrimp lethality assay ($LC_{50}$ values) of ethyl acetate extracts from the investigated fungal isolates.

| Fungal Isolate AUMC No. | $IC_{50}$ Value (mg/mL) | $LC_{50}$ Value (mg/mL) |
|---|---|---|
| 15440 | 3.51 ± 0.02 [b] | 1.18 |
| 15443 | 42.87 ± 1.73 [f] | 2.01 |
| 15441 | 17.69 ± 0.07 [e] | 1.45 |
| 15446 | 9.73 ± 0.01 [d] | 1.34 |
| 15444 | 6.06 ± 0.10 [c] | 1.39 |
| 15447 | 0.47 ± 0.00 [a] | 2.00 |
| 15448 | 0.58 ± 0.01 [a] | 1.29 |
| 15445 | 11.09 ± 0.01 [d] | 1.56 |
| 15442 | 17.12 ± 0.19 [e] | 1.71 |
| BHT | 3.38 ± 0.04 [b] | - |

Data are presented as the mean ± SD, and values associated with superscripts differ significantly at $p < 0.05$.

### 3.5. Evaluation of Total Phenolics and Flavonoids

The results proved that the *A. terreus* AUMC 15447 EtOAc extract had the highest content of phenolics (138.30 mg/g) and flavonoids (72.09 mg/g), while the *A. terreus* AUMC 15448 EtOAc extract showed good phenolic and flavonoid content (116.54 and 53.60 mg/g, respectively). The lowest content of phenolics was noted in the extracts of *P. novae-zeelandiae* AUMC 15442 and *T. harzianum* AUMC 15443 (20.60 and 25.81 mg/g, respectively), and the lowest flavonoid content in *A. aureolatus* AUMC 15441 and *P. novae-zeelandiae* AUMC 15442 (17.43 and 18.48 mg/g, respectively) (Table 7).

**Table 7.** Total phenolic and flavonoid content of ethyl acetate extracts from the investigated fungal isolates.

| Fungal Isolate AUMC No. | Total Phenolics Gallic Acid Equivalent (mg/g) | Total Flavonoids Quercetin Equivalent (mg/g) |
|---|---|---|
| 15440 | 38.95 ± 1.45 [e] | 32.14 ± 0.40 [c] |
| 15443 | 25.81 ± 0.52 [g] | 22.94 ± 0.65 [e] |
| 15441 | 31.87 ± 0.70 [f] | 17.43 ± 0.40 [g] |
| 15446 | 103.11 ± 0.71 [c] | 29.09 ± 0.50 [d] |
| 15444 | 51.86 ± 1.01 [d] | 21.66 ± 0.39 [e,f] |
| 15447 | 138.30 ± 1.14 [a] | 72.09 ± 0.71 [a] |
| 15448 | 116.54 ± 1.04 [b] | 53.60 ± 0.49 [b] |
| 15445 | 28.04 ± 0.85 [g] | 20.42 ± 0.42 [f] |
| 15442 | 20.60 ± 0.26 [h] | 18.48 ± 0.17 [g] |

Data are presented as the mean ± SD, and values associated with superscripts differ significantly at $p < 0.05$.

### 3.6. Flavonoid and Phenolic HPLC Profile of Extracts

HPLC analysis for profiling the flavonoid and phenolic chemical constituents of EtOAc extracts of the derived fungal isolates was performed. Standards utilized in the analysis, according to their retention time, included chlorogenic acid, gallic acid, catechin, methyl gallate, caffeic acid, syringic acid, rutin, ellagic acid, *p*-coumaric acid, vanillin, cinnamic acid, ferulic acid, daidzein, naringenin, quercetin, apigenin, kaempferol, and hesperetin, as shown in Table 8 and Supplementary Figures S1–S9.

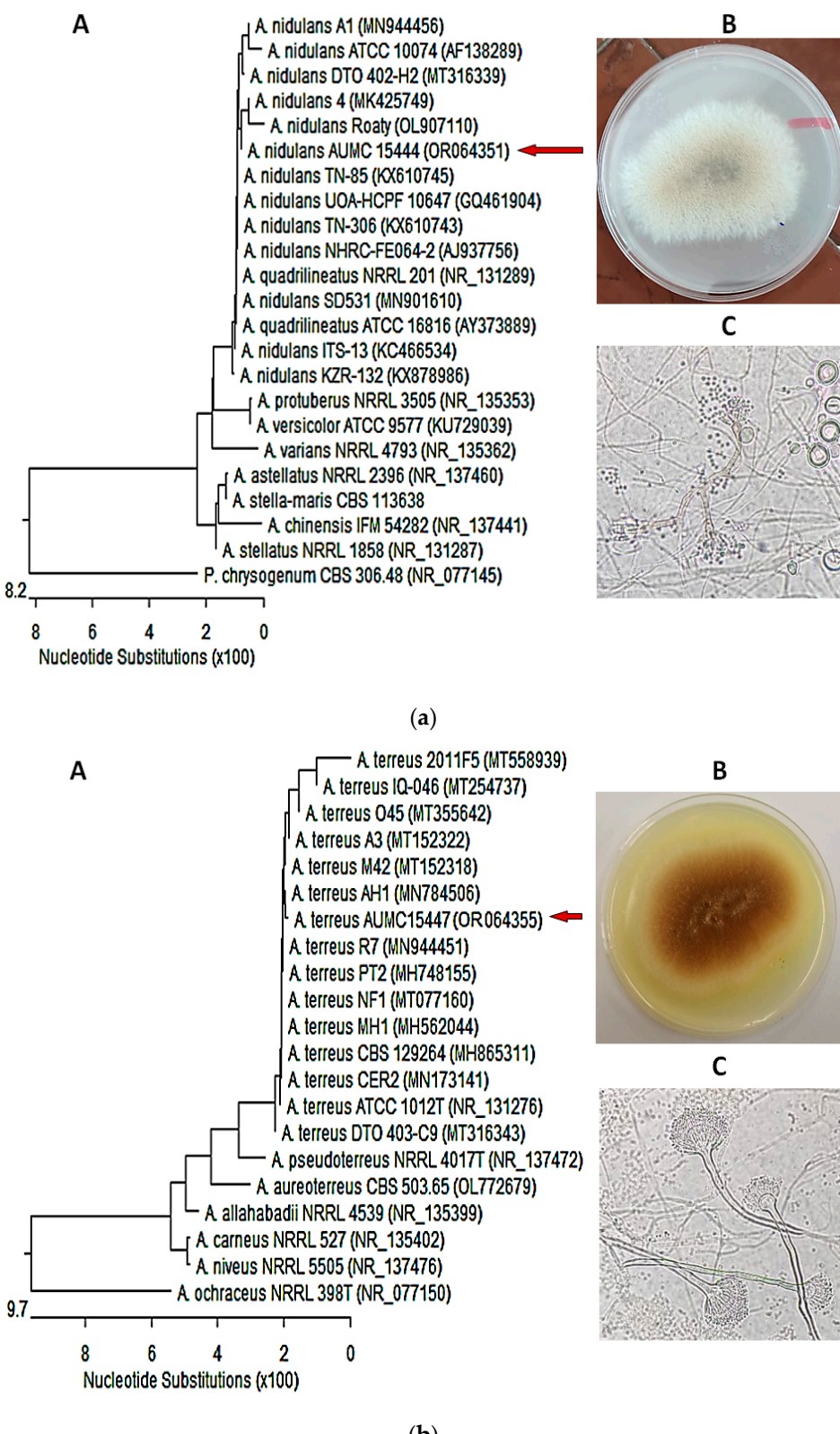

**Figure 1.** (**a**) *Aspergillus nidulans* AUMC 15444: (**A**) Phylogenetic tree based on the ITS sequencing of rDNA; (**B,C**) Culture characteristics, microscope's magnification 400×. (**b**) *Aspergillus terreus* AUMC 15447: (**A**) Phylogenetic tree based on the ITS sequencing of rDNA; (**B,C**) Culture characteristics, microscope's magnification 400×.

**Table 8.** Retention time ($R_t$), area percentage, and concentration (µg/g) of HPLC-detected phenolic compounds from EtOAc fungal isolate extracts.

| Peak | RT (min) | Name | AUMC 15440 Area % | Conc. (µg/g) | AUMC 15443 Area % | Conc. (µg/g) | AUMC 15441 Area % | Conc. (µg/g) | AUMC 15446 Area % | Conc. (µg/g) | AUMC 15444 Area % | Conc. (µg/g) | AUMC 15447 Area % | Conc. (µg/g) | AUMC 15448 Area % | Conc. (µg/g) | AUMC 15445 Area % | Conc. (µg/g) | AUMC 15442 Area % | Conc. (µg/g) |
|---|---|---|---|---|---|---|---|---|---|---|---|---|---|---|---|---|---|---|---|---|
| 1 | 3.49 | Gallic acid | 2.07 | 410.40 | 3.62 | 142.75 | 0.80 | 115.12 | 0.56 | 635.79 | 0.46 | 42.13 | 0.24 | 140.54 | 2.18 | 680.12 | 1.31 | 85.8 | 1.76 | 108.74 |
| 2 | 4.38 | Chlorogenic acid | 6.26 | 1937.16 | 27.44 | 1685.53 | 66.39 | 14,915.62 | 23.36 | 41,284.66 | 3.64 | 522.12 | 0.36 | 331.10 | 9.49 | 4614.26 | 40.99 | 4167.3 | 4.50 | 432.88 |
| 3 | 4.74 | Catechin | 0.00 | ND | 0.26 | 17.90 | 0.00 | ND | 0.00 | ND | 0.15 | 24.37 | 0.00 | ND | 0.00 | ND | 0.00 | ND | 0.00 | ND |
| 4 | 5.68 | Methyl gallate | 13.90 | 372.12 | 19.89 | 105.73 | 0.23 | 4.49 | 0.14 | 20.71 | 3.40 | 42.19 | 0.26 | 20.58 | 0.25 | 10.68 | 0.43 | 3.8 | 0.24 | 1.99 |
| 5 | 6.17 | Caffeic acid | 13.26 | 1969.90 | 11.10 | 327.34 | 1.11 | 119.96 | 2.78 | 2359.24 | 15.43 | 1062.60 | 0.03 | 15.07 | 3.02 | 704.88 | 10.18 | 497.0 | 1.27 | 58.73 |
| 6 | 6.66 | Syringic acid | 0.08 | 10.23 | 1.54 | 39.77 | 9.20 | 867.33 | 14.49 | 10,746.51 | 2.85 | 171.22 | 0.05 | 20.85 | 3.74 | 762.56 | 0.82 | 34.9 | 1.43 | 57.79 |
| 7 | 8.21 | Rutin | 0.03 | 10.64 | 8.17 | 553.19 | 0.78 | 194.25 | 5.11 | 9946.40 | 1.88 | 297.56 | 0.33 | 338.60 | 3.64 | 1949.47 | 4.52 | 506.1 | 0.47 | 49.32 |
| 8 | 9.15 | Ellagic acid | 12.34 | 3586.82 | 4.17 | 240.92 | 2.31 | 486.81 | 2.40 | 3977.62 | 1.26 | 169.33 | 0.10 | 86.16 | 1.60 | 731.22 | 1.07 | 102.5 | 0.19 | 17.20 |
| 9 | 9.24 | *p*-Coumaric acid | 1.02 | 56.50 | 2.73 | 30.13 | 0.41 | 16.41 | 0.02 | 6.56 | 1.08 | 27.73 | 0.04 | 6.18 | 0.64 | 55.79 | 0.86 | 15.7 | 0.37 | 6.31 |
| 10 | 9.76 | Vanillin | 0.91 | 69.33 | 0.24 | 3.59 | 0.47 | 25.87 | 0.45 | 196.28 | 0.10 | 3.51 | 0.00 | ND | 0.52 | 62.81 | 0.09 | 2.4 | 0.42 | 9.92 |
| 11 | 10.32 | Ferulic acid | 11.37 | 1544.83 | 5.14 | 138.74 | 2.25 | 222.14 | 2.27 | 1763.41 | 3.89 | 245.23 | 1.47 | 596.68 | 14.19 | 3029.11 | 1.96 | 87.6 | 3.15 | 133.19 |
| 12 | 10.58 | Naringenin | 0.73 | 167.46 | 1.39 | 63.46 | 1.04 | 173.17 | 1.63 | 2145.88 | 1.19 | 127.38 | 2.16 | 1490.00 | 7.50 | 2713.25 | 0.39 | 29.7 | 2.73 | 195.45 |
| 13 | 12.33 | Daidzein | 2.12 | 226.28 | 4.52 | 96.05 | 6.96 | 540.72 | 28.24 | 17,248.31 | 8.05 | 398.89 | 23.18 | 7426.39 | 5.66 | 950.92 | 1.83 | 64.3 | 7.36 | 244.70 |
| 14 | 12.81 | Quercetin | 0.72 | 121.88 | 0.65 | 22.05 | 0.79 | 97.93 | 3.10 | 3010.94 | 14.78 | 1165.77 | 26.30 | 13,411.96 | 1.12 | 300.74 | 0.27 | 15.2 | 3.54 | 187.42 |
| 15 | 14.03 | Cinnamic acid | 0.34 | 11.14 | 5.13 | 33.21 | 1.68 | 39.68 | 0.37 | 69.20 | 1.49 | 22.51 | 41.65 | 4070.05 | 36.58 | 1874.54 | 0.81 | 8.7 | 64.83 | 657.33 |
| 16 | 14.54 | Apigenin | 15.30 | 2678.13 | 0.53 | 18.57 | 4.18 | 531.59 | 8.67 | 8669.62 | 10.41 | 844.85 | 3.38 | 1770.67 | 4.06 | 1116.32 | 13.52 | 777.5 | 0.00 | ND |
| 17 | 15.03 | Kaempferol | 0.04 | 6.85 | 1.61 | 59.46 | 0.00 | ND | 0.00 | ND | 0.69 | 59.40 | 0.28 | 153.69 | 1.24 | 360.20 | 9.34 | 569.5 | 0.00 | ND |
| 18 | 15.55 | Hesperetin | 0.38 | 31.35 | 1.85 | 30.25 | 1.40 | 83.56 | 6.41 | 3014.19 | 29.24 | 1116.63 | 0.19 | 47.99 | 4.56 | 590.43 | 11.60 | 314.0 | 7.74 | 198.38 |

ND, Not detected.

Ellagic acid, vanillin, and apigenin were present at high levels in *T. harzianum* AUMC 15440 (12.34, 0.91, and 15.30%, respectively) in comparison with other isolate extracts, with apigenin being a major compound. The *Trichoderma harzianum* AUMC 15443 extract contained gallic acid (3.62%), catechin (0.26%), methyl gallate (19.89%), rutin (8.17%), and *p*-coumaric acid (2.73%) in high concentrations compared with other extracts, with methyl gallate being a major constituent. Chlorogenic acid (66.39%) was detected at high levels and was a major phenolic in *A. aureolatus* AUMC 15441, while syringic acid and daidzein, as major chemical constituents, were present in large quantities (14.49 and 28.24%) in the *A. aureolatus* AUMC 15441 extract. Meanwhile, elevated quantities of caffeic acid (15.43%) and hesperetin (29.24%) as the major components in the *A. nidulans* AUMC 15444 extract, quercetin (26.30%) in *A. terreus* AUMC 15447, as well as ferulic acid and naringenin (14.19% and 7.50%, respectively) in the *A. terreus* AUMC 15448 EtOAc extract were detected. Among all the extracts, the highest concentrations of kaempferol (9.34%) and cinnamic acid (64.83%) as the major phenolics were detected in extracts derived from *P. crustosum* AUMC 15445 and *P. novae-zeelandiae* AUMC 15442, respectively. The results of the detected phenolics revealed that cinnamic acid (41.65 and 36.58%) and chlorogenic acid (40.99%) are the major phenolics quantified in *A. terreus* AUMC 15447 and AUMC 15448 and *P. crustosum* AUMC 15445 extracts, respectively. On the other hand, neither vanillin nor apigenin were detected in *A. terreus* AUMC 15447 and *P. novae-zeelandiae* AUMC 15442 extracts, while kaempferol was not detected in *A. aureolatus* AUMC 15441and AUMC 15446 and *P. novae-zeelandiae* AUMC 15442 extracts. Catechin was detected only in *T. harzianum* AUMC 15443 and *A. nidulans* AUMC 15444 extracts.

### 3.7. In Silico Molecular Docking of Identified Compounds

According to the HPLC analysis, the diverse extracts were found to contain approximately 18 phenolic compounds, as detailed in Table 8. Docking studies were then conducted to assess the binding affinities of these compounds as potential antimicrobials against both *S. aureus* and *C. albicans*. The ranking of binding interactions for each compound with the respective target proteins was based on factors such as the lowest energy and lowest RMSD (Root-Mean-Square Deviation). It is noteworthy that a lower binding energy score indicates better stability in the protein–ligand binding interaction [80].

### 3.7.1. Molecular Docking Simulation against *S. aureus* Tyrosyl-tRNA Synthetase

The exploration of molecular simulation outcomes for the identified molecules against *S. aureus* tyrosyl-tRNA synthetase indicated diverse affinities for the enzyme, ranging from −16.43 kcal/mol (rutin) to −7.30 kcal/mol (cinnamic acid), suggesting their potential as inhibitors (Table 9). Rutin, chlorogenic acid, and ellagic acid emerged as the top-scoring compounds, exhibiting pose scores of −16.43 (RMSD = 1.32 Å), −13.94 (RMSD = 1.40 Å), and −13.89 (RMSD = 0.55 Å) kcal/mol, respectively. Notably, the highest-affinity compound, rutin, demonstrated hydrogen-bond interactions with Cys 37 and Asp 40 amino acid residues, as revealed in Figure 2. Additionally, the 2D and 3D interaction models illustrated the involvement of hydrophobic interactions with the Ala 39, Pro 53, and Phe 54 amino acid residues. Also, the interactions of rutin with the enzyme include an acidic interaction with Asp 40, 80, and 195 as well as basic interactions with the Lys 84 and Arg 88 amino acid residues. Other polar interactions for chlorogenic and ellagic acids are shown in Figures 3 and 4.

**Table 9.** Pose score results of detected compound interactions with target proteins.

| Seq. | Compound | *S. aureus* Tyrosyl-tRNA Synthetase | | *C. albicans* Aspartic Protease 2 | |
|---|---|---|---|---|---|
| | | Score (kcal/mol) | RMSD (Å) | Score (kcal/mol) | RMSD (Å) |
| 1 | Caffeic acid | −12.57 | 0.58 | −7.71 | 1.12 |
| 2 | Catechin | −11.10 | 0.96 | −9.93 | 1.03 |
| 3 | Chlorogenic acid | −13.94 | 1.40 | −9.57 | 1.34 |
| 4 | Cinnamic acid | −7.30 | 0.98 | −9.08 | 0.84 |
| 5 | Ellagic acid | −13.89 | 0.55 | −9.09 | 0.67 |
| 6 | Ferulic acid | −9.04 | 0.60 | −11.15 | 0.60 |
| 7 | Gallic acid | −10.06 | 0.70 | −9.64 | 0.52 |
| 8 | Hesperetin | −11.17 | 0.80 | −8.47 | 0.92 |
| 9 | Kaempferol | −11.76 | 0.89 | −10.97 | 0.72 |
| 10 | Methyl gallate | −11.47 | 0.66 | −10.76 | 0.77 |
| 11 | Naringenin | −11.45 | 0.68 | −9.21 | 0.67 |
| 12 | *p*-Coumaric acid | −7.97 | 1.00 | −7.82 | 1.03 |
| 13 | Querectin | −12.76 | 0.78 | −9.30 | 1.03 |
| 14 | Rutin | −16.43 | 1.32 | −12.35 | 1.05 |
| 15 | Syringic acid | −10.08 | 0.63 | −8.91 | 1.25 |
| 16 | Vanillin | −8.03 | 0.65 | −6.33 | 1.06 |
| 17 | Apigenin | −12.44 | 0.85 | −8.71 | 0.90 |
| 18 | Daidzein | −10.53 | 0.64 | −7.86 | 1.00 |

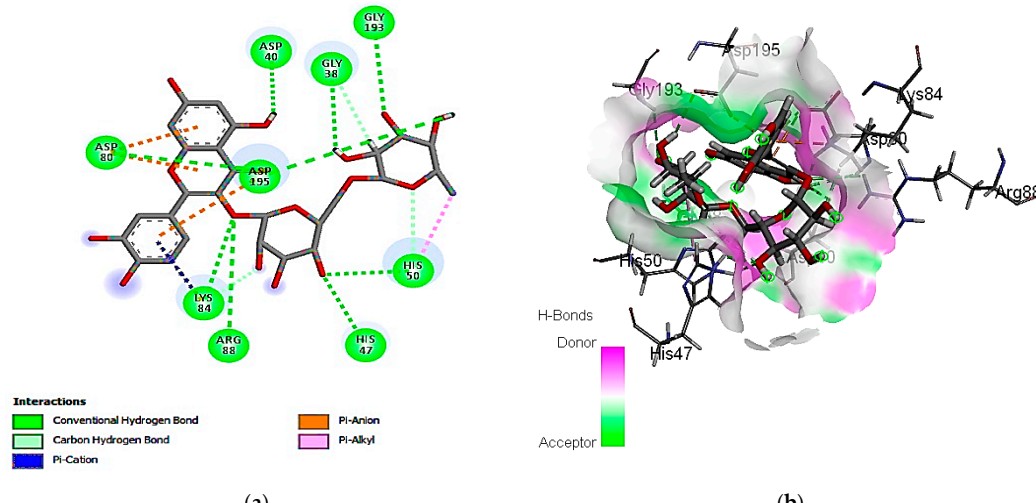

(**a**)　　　　　　　　　　　　(**b**)

**Figure 2.** Predicted binding-pose interactions of rutin with residues of the *S. aureus* tyrosyl-tRNA synthetase target. (**a**) 2D interaction model; (**b**) 3D interaction model.

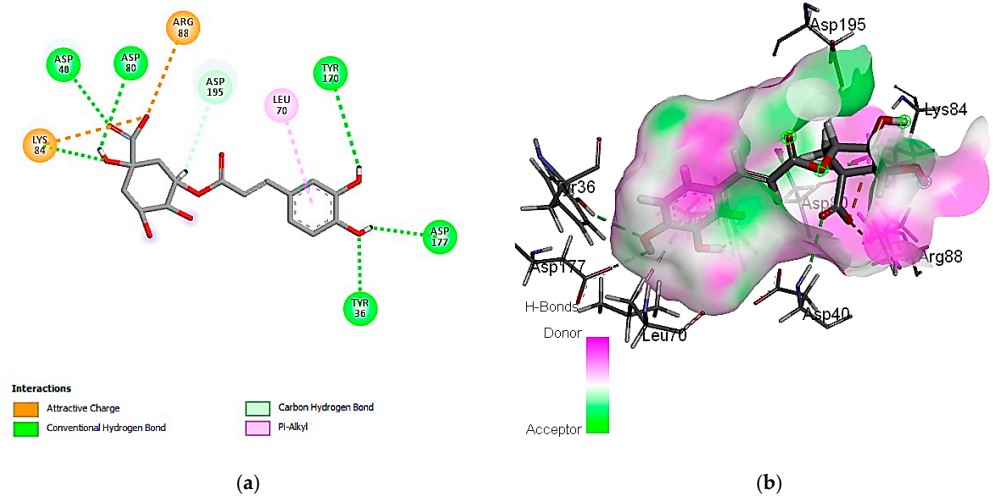

(**a**)　　　　　　　　　　　　(**b**)

**Figure 3.** Predicted binding-pose interactions of chlorogenic acid with residues of the *S. aureus* tyrosyl-tRNA synthetase target. (**a**) 2D interaction model; (**b**) 3D interaction model.

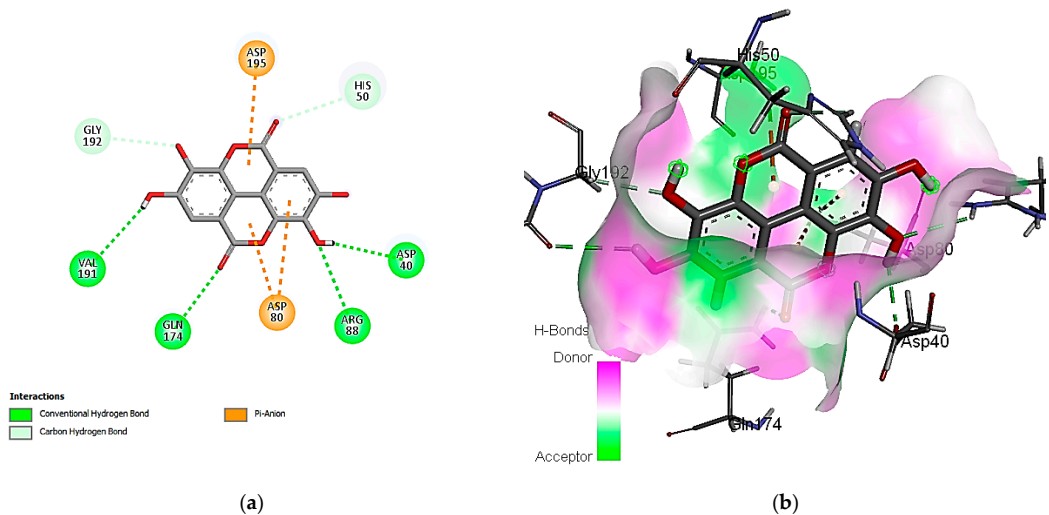

(**a**)                                    (**b**)

**Figure 4.** Predicted binding-pose interactions of ellagic acid with residues of the *S. aureus* tyrosyl-tRNA synthetase target. (**a**) 2D interaction model; (**b**) 3D interaction model.

3.7.2. Molecular Docking Simulation with the *C. albicans* Secreted Aspartic Protease 2

Concerning the binding affinity of the investigated compounds against the *C. albicans* secreted aspartic protease 2 (Table 9), the compounds rutin, ferulic acid, and kaempferol achieved the highest binding energy scores of $-12.3505$, $-11.1575$, and $-10.9706$ kcal/mol, respectively. The binding mode of rutin showed HB binding with the Asp 86, Asp 120, and Thr 222 amino acid residues (Figure 5). It is noteworthy that hydrophobic interactions are involved in the binding of rutin with the Ile 30, 119, 123, and Val 12 amino acid residues. Other polar interactions for ferulic acid and kaempferol are shown in Figures 6 and 7.

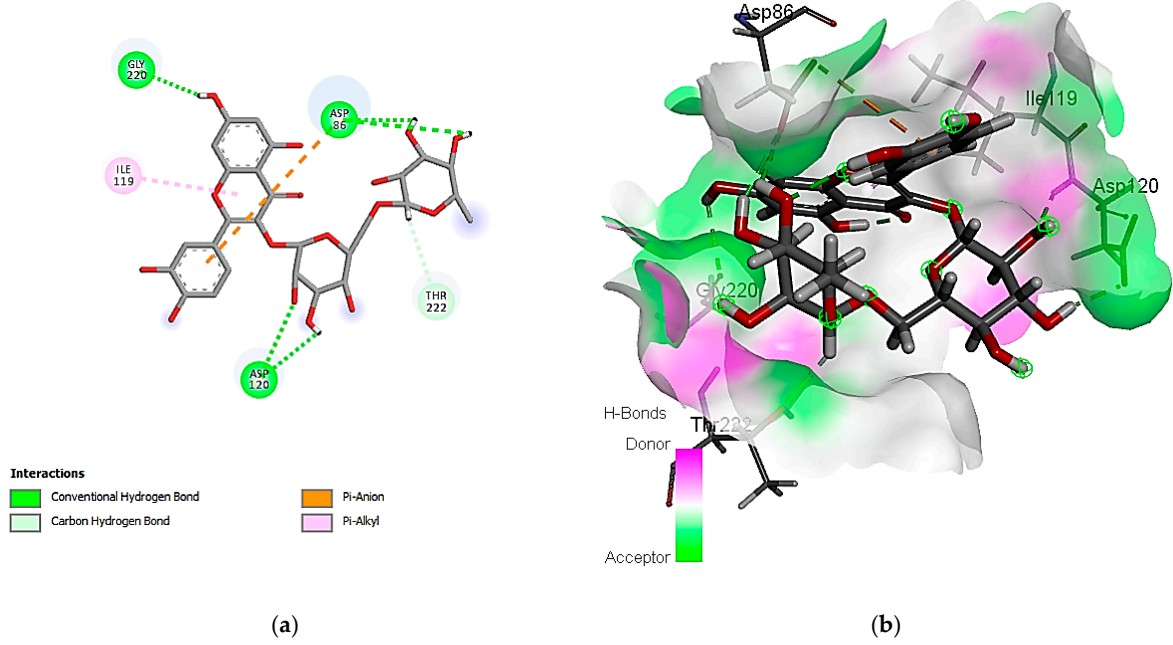

(**a**)                                    (**b**)

**Figure 5.** Predicted binding-pose interactions of rutin with residues of the *C. albicans* secreted aspartic protease 2. (**a**) 2D interaction model; (**b**) 3D interaction model.

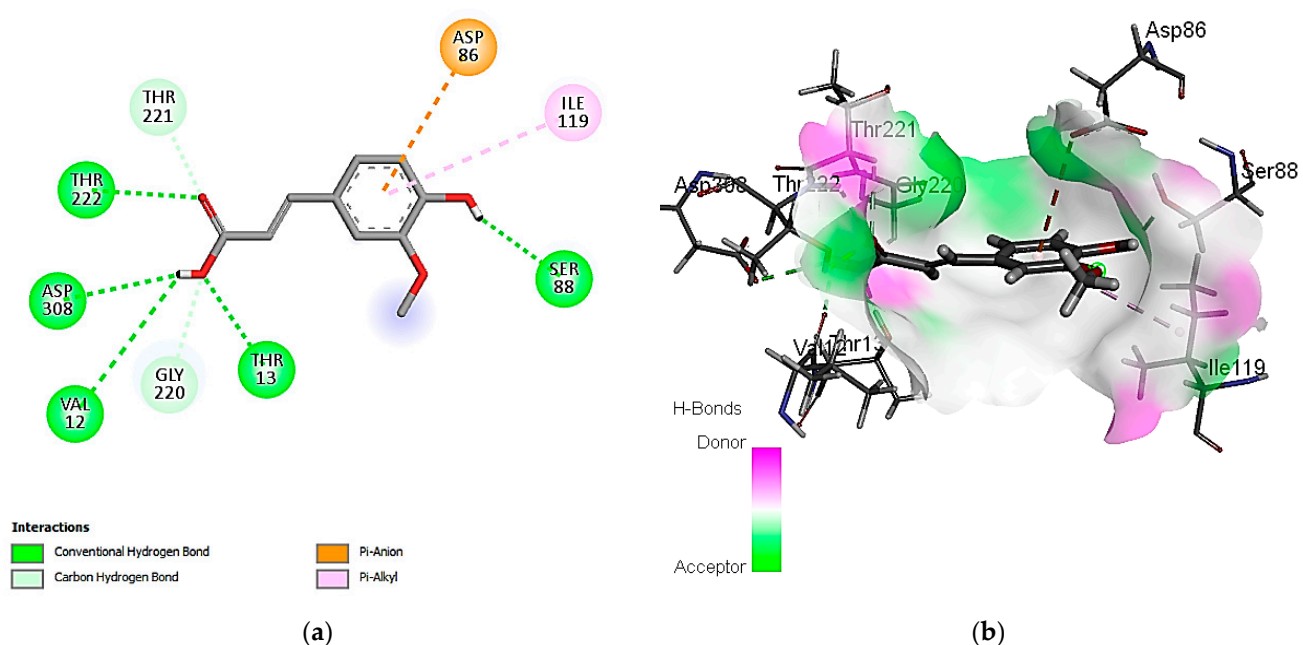

**Figure 6.** Predicted binding-pose interactions of ferulic acid with residues of the *C. albicans* secreted aspartic protease 2. (**a**) 2D interaction model; (**b**) 3D interaction model.

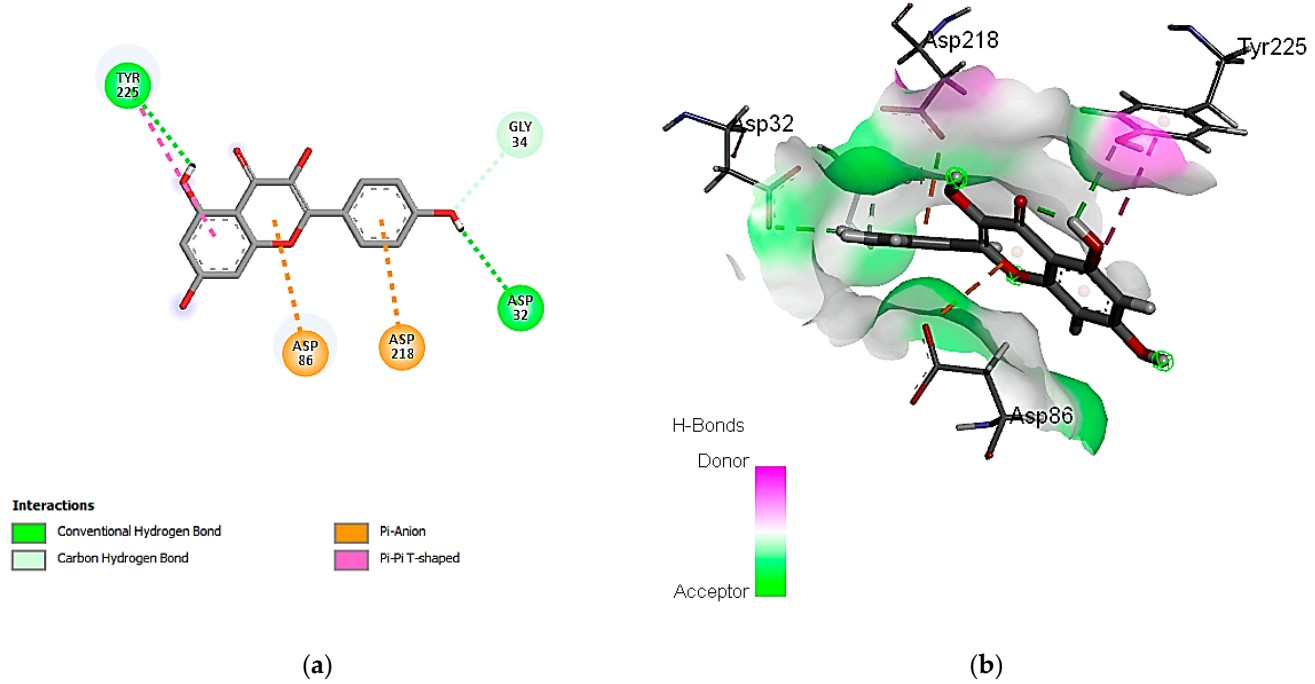

**Figure 7.** Predicted binding-pose interactions of kaempferol with residues of the *C. albicans* secreted aspartic protease 2. (**a**) 2D interaction model; (**b**) 3D interaction model.

## 4. Discussion

Fungi are a prolific source of bio-effective secondary metabolites, including those with antimicrobial efficiency, which have been developed into significant pharmaceuticals. Endophytic fungi have been documented as having the capacity to generate novel antibacterial, antifungal, antiviral, anti-inflammatory, antitumor, and antimalarial compounds. These bioactive natural products are eliciting substantial interest from biologists and natural product chemists [81]. Numerous diseases caused by *S. aureus*, ranging from mild skin

infections to more serious and potentially fatal diseases, accompanied by the emergence of multi-resistant strains, have increased the demand for new antibacterial agents [53]. Limited availability, in addition to the side effects of efficient antifungal agents, are some of the issues faced with *C. albicans* infections [82]. Consequently, it is imperative to develop new drugs by exploring alternate sources such as terrestrial and marine environments for novel medications [17,83].

In the current investigation, nine EtOAc extracts were obtained from soil fungal isolates belonging to six different fungal species, namely *T. harzianum*, *A. aureolatus*, *A. nidulans*, *A. terreus*, *P. crustosum*, and *P. novae-zeelandiae*, which exerted anti-staphylococcal and anti-candidal influences against tested species of *Staphylococcus* and *Candida*. The resultant anti-staphylococcal activities of the different fungal extracts at 100 mg/mL revealed that most of these extracts exhibited efficiency against the tested *S. aureus* species. *Aspergillus terreus*, *A. nidulans*, and *T. harzianum* extracts exhibited the highest activity, with MIC values ranging between 25 and 0.39 mg/mL. On the other hand, the *A. nidulans* extract revealed a high potency, with a MIC of 3.125–0.39 mg/mL against *Candida* spp. The *Aspergillus terreus* MK-1 ethyl acetate extract showed antagonistic activity against *S. aureus* and *C. albicans*, with inhibition zones of 45 and 30 mm, respectively [84]. Phupiewkham et al. [85] reported that *T. harzianum* TS3 and TS12 filtrates exert a positive control effect against *S. aureus* ATCC25923. A crude extract from *T. harzianum* exhibited the highest zone of inhibition of 26 mm against methicillin-resistant *S. aureus* [86]. A *Trichoderma harzianum* extract had an MIC of 50 μg/mL against *S. aureus* [87]. An *Aspergillus nidulans* ethyl acetate extract against an oral isolate of *C. albicans* (CA 09) showed a growth inhibitory effect, with an 18.4 mm diameter inhibition zone and an MIC value of 250 μg/mL [88]. External secondary metabolites in an *n*-butanol extract from *Aspergillus terreus* var. *africans* exhibited a 22 and 19 mm diameter inhibition zone against *C. albicans* and *C. glabrata*, respectively [12]. The ability of *C. albicans* to transform into a hyphal state and adhere to tissue surfaces is one of the main causes for its severe pathogenicity [88].

*Staphylococcus aureus* exerted resistance to gentamicin and chloramphenicol while presenting susceptibility to amoxicillin and vancomycin, with MICs of 0.1 and 0.015 mg/mL, respectively [89]. Antibacterial agents such as ampicillin, amoxicillin, chloramphenicol, and cefotaxime had MIC values of 39, 39, 16, and 1 μg/mL, respectively, against *S. aureus* ATCC 11632 [90]. However, ampicillin exhibited MICs of 2.5 and 10 μg/mL on *S. aureus* ATCC 25923 and *S. aureus* TN2A, respectively [52]. Al Halteet et al. [65], in their study to determine the antifungal susceptibility of 24 *Candida* species, indicated that *C. krusei* MN419370 exhibited resistance to all the tested antifungal agents (fluconazole, voriconazole, caspofungin, micafungin, amphotericin B, and flucytosine), while *C. krusei* MN419388 was resistant only to fluconazole and flucytosine. Meanwhile, 2 out of 18 *C. albicans* strains showed resistance to all antifungal agents. On the other hand, three *C. glabrata* strains were susceptible to all tested antifungal agents.

The utilization of the microbroth dilution method using microtiter plates has emerged as the technique of choice for assessing drug susceptibility testing due to its low sample needs, affordability, and high throughput rate [91]. The *p*-iodonitrotetrazolium (INT) was chosen as an indicator because of its preparation in ethanol that, as an antiseptic, makes it appropriate for reducing the contamination risk during test procedures [92]. Violet INT is a tetrazolium dye precursor that, upon reduction, constitutes a purple formazan whose intensity is directly proportional to the number of live cells. The reduction of INT is due to the mitochondrial enzymatic activity within living cells [67].

According to the results, the *A. terreus* AUMC 15447 extract exhibited the highest phenolic content of 138.30 mg/g, a flavonoid content of 72.09 mg/g, and an $IC_{50}$ of 0.47 mg/mL by the anti-DPPH radical-scavenging assay, followed by the *A. terreus* AUMC 15448 extract. The ethyl acetate extract fraction from *A. terreus* LS01 showed an $IC_{50}$ of 19.91 μg/mL in the in vitro anti-DPPH radical-scavenging assay [93]. A total phenolic content of 122.475 mg/g was detected in an *A. terreus*-18 EtOAc extract with the highest DPPH scavenging activity of 80.4% inhibition [94]. Chandra and Arora [95] determined a phenolic content (20.4

and 16.7 mg/mL) and DPPH scavenging efficiency (86.8 and 65.8%) in *A. terreus*-1 and -2 soil isolates, respectively, after fermentation on sugarcane bagasse. *Aspergillus nidulans* ST22 showed total phenolic and flavonoid amounts of 0.1413 and 0.01162 mg/mL, respectively, and a 34.17% antioxidative capacity by a hydroxyl radical-scavenging assay [96]. Goncalves and Pombeiro-Sponchiado [97] reported the ability of melanin from *A. nidulans* MEL1 to scavenge the oxidants HOCl and $H_2O_2$. Al-Askar [98] established total phenols of 53.30 µg/mL in a *T. harzianum* 1-SSR filtrate. Tavares et al. [99] evaluated a TPC of 8 mg/g and an $EC_{50}$ value of 25.41 µg/mL for free radical DPPH in an ethyl acetate extract of *A. aureolatus* CML2964 isolated from caves, while Ahmed and Al-Shamary [100] determined a phenolic content of 51 µg/mL in the soil *A. niger* B1b strain. In our results, *T. harzianum* AUMC 15440 recorded an $LC_{50}$ of 1.18 mg/mL in the brine- shrimp cytotoxicity assay, while the *A. terreus* AUMC 15448 extract showed an $LC_{50}$ of 1.29 mg/mL. The biotoxicity of the *A. terreus* var. *africanus* crude ethyl acetate extract showed a slight toxicity against *Artemia salina* ($LC_{50}$ = 1500 µg/mL) [101].

The most effective fungal isolates were subjected to phylogenetic analysis by ITS region sequencing, which molecularly identified them as *A. nidulans* AUMC 15444 and *A. terreus* AUMC 15447. The ITS regions represent important efficient markers for confirming fungal strain identification at the species level [102]. Molecular identification of *Aspergillus* strains at the species level using ITS region sequencing was presented as an efficient substitute approach for their accurate identification [103].

HPLC analysis for the detection of phenolic compounds in fungal EtOAc extracts with reference standards showed great variation in the types and quantities of detected constituents among the fungal extracts. In the same context, Hassane et al. [10,11] detected a variety of phenolic and flavonoid compounds in fungal extracts belonging to different genera of *Aspergillus*, *Penicillium*, *Chaetomium*, and *Rhizopus*. Abdel-Wareth and Ghareeb [104] identified 22 phenolic and flavonoid compounds (cinnamic acid, 3,4,5-methoxy cinnamic acid, α-coumaric acid, *p*-coumaric acid, coumarin, pyrogallol, gallic acid, methyl gallate, ellagic acid, benzoic acid, 4-amino-benzoic acid, catechin, epicatechin, protocatechuic acid, vanillic acid, ferulic acid, isoferulic acid, chlorogenic acid, caffeic acid, resveratrol, and salicylic acid) in *Penicillium implicatum* and *Aspergillus niveus* extracts from fresh-water isolates by RP-HPLC/DAD analysis, where pyrogallol and *e*-vanillic were the major phenolics detected in the extracts, respectively. Abdel-Aty et al. [105] optimized the production of *p*-coumaric, apigenin, and kaempferol using SSF of *Trichoderma reesei* with a TPC of 30 mg/g. The diorcinol phenolic was detected and isolated from *A. nidulans* MCCC 3A00050 [106]. Shevelev et al. [44] reported the in vivo efficacy of polyphenols, including resveratrol, dihydroquercetin, and dihydromyricetin, in the treatment of infected wounds with *S. aureus* ATCC 25923 and *C. albicans* NCTC 2625. Kepa et al. [107] demonstrated the promised efficiency of caffeic acid with MICs (256–1024 µg/mL) against MR *S. aureus*, clinical isolates, and reference strains derived from wound infections.

Moreover, advancements in computational chemistry have enabled the development of some fungal-derived anti-pathogenic compounds with potential therapeutic influences in controlling health-threatening microbes [53]. The docking results of detected phenolics against the *S. aureus* tyrosyl-tRNA synthetase and the *C. albicans* secreted aspartic protease 2 found that rutin had the lower binding energy score, which conferred on it better protein–ligand binding stability [108]. Rutin and rutin–gentamicin act as pro-oxidants against *Pseudomonas aeruginosa* through oxidative stress by inducing reactive oxygen species generation, which leads to cell death [109]. Docking of flavonoids—saltillin, taxifolin, and 6-methoxyflavone—from the *A. nidulans* chloroform extract displayed good binding interactions with the *C. albicans* growth regulator N-myristoyltransferase [88].

Naturally derived substances are a prime contender for treating microbial infections and restricting the spread of drug-resistant strains because of their capacity to interact with the microbial cell through a variety of antimicrobial processes [110,111]. Phenolic acids and their derivatives showed antimicrobial effects by influencing the solubility of microbial membranes, rupturing the cellular membranes and allowing vital intracellular

components to outflow, thus resulting in death of the microbial cell [23,112,113]. The compounds' capacity to pass through the cytoplasmic membrane and acidify the cytoplasm is dependent on the number of hydroxyl, methoxy, and carboxyl groups in addition to the saturation state of the alkyl side chain [114]. Lou et al. [115] evaluated *p*-coumaric acid's antimicrobial efficacy in triggering bacterial cell death by disturbing the cell membrane and the interaction with genomic DNA. Wu et al. [26] reported that flavonoids and their isomers represent a valuable anti-pathogenic mediator by targeting a variety of pathogens and suppressing virulence properties in drug-resistant microbial strains. Their capacity to impede microbial cell energy metabolism, nucleic acid synthesis, complex formation with the bacterial cell wall, and cytoplasmic membrane function, are the mechanisms linked to their bacteriostatic actions [25,116].

## 5. Conclusions

Ethyl acetate (EtOAc) extracts from the fungal isolates exhibited notable anti-staphylococcal and anti-candidal efficacy, with *A. terreus* AUMC 15447 and *A. nidulans* AUMC 15444 strains demonstrating particularly high efficiency, as determined through microdilution assays combined with INT for assessing antimicrobial efficacy. Additionally, the assayed extracts displayed considerable phenolic and flavonoid contents, along with significant $IC_{50}$ values for antioxidative activity and $LC_{50}$ values in *Artemia* cytotoxicity assays. HPLC analysis revealed the presence of various phenolics in different concentrations within the fungal EtOAc extracts. Molecular simulations of these detected phenolics highlighted the high affinity of rutin with the *S. aureus* tyrosyl-tRNA synthetase and the *C. albicans* secreted aspartic protease 2, indicating inhibitory properties. This research suggests the potential for bioprospecting anti-staphylococcal and anti-candidal bio-active metabolites from fungi, providing a sustainable source for safe medication.

**Supplementary Materials:** The following supporting information can be downloaded at: https://www.mdpi.com/article/10.3390/cimb46010016/s1, HPLC chromatograms of phenolic components in EtOAc extracts (Figures S1–S9).

**Author Contributions:** Conceptualization, A.M.A.H. and A.A.A.M.; methodology, M.E.A., A.M.A.H. and Y.A.-D.; software, M.E.A. and Y.A.-D.; validation, A.A.A.M., M.E.A. and N.S.A.G.; formal analysis, H.M.; investigation, M.E.A. and H.M.; resources, M.E.A. and A.M.A.H.; data curation, H.M.; writing—original draft preparation, H.M., N.S.A.G. and Y.A.-D.; writing—review and editing, M.E.A., A.A.A.M. and A.M.A.H.; visualization, A.M.A.H. and N.F.A.-D.; supervision, A.A.A.M. and A.M.A.H.; project administration, A.A.A.M. and N.F.A.-D.; funding acquisition, A.A.A.M. All authors have read and agreed to the published version of the manuscript.

**Funding:** This research was funded by the Distinguished Scientist Fellowship Program through the Researchers Supporting Project (RSP2023R227), King Saud University, Riyadh, Saudi Arabia.

**Institutional Review Board Statement:** Not applicable.

**Informed Consent Statement:** Not applicable.

**Data Availability Statement:** Data are contained within the article and Supplementary Materials.

**Conflicts of Interest:** The authors declare no conflicts of interest.

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
