# Peer review of "Anti-Staphylococcal, Anti-Candida, and Free-Radical Scavenging Potential of Soil Fungal Metabolites: A Study Supported by Phenolic Characterization and Molecular Docking Analysis"

_cimb, doi:10.3390/cimb46010016_

Round 1
Reviewer 1 Report
Comments and Suggestions for Authors
Staphylococcus and Candida are acknowledged as etiological agents in a multitude of diseases, and the increasing prevalence of multidrug-resistant strains emphasizes the need to investigate natural sources, especially soil-fungi, as potential sources of antimicrobial agents. This study explores and demonstrates the remarkable efficacy of the antimicrobial potential of ethyl acetate extracts of soil-derived fungi against Staphylococcus and Candida. Noteworthy levels of efficiency were observed, especially in the case of the A. terreus AUMC 15447 and A. nidulans AUMC 15444 strains. Complementary analyzes and determinations support the authors' theory. The paper is interesting and multidisciplinary. The article was well documented and written, presenting a topic relevant to academia. I take this opportunity to congratulate the authors for their complex, excellent work. The article is interesting, however, there are small errors in the manuscript. However, in present form several explanations and modifications are needed. Overall experiments are well done, and the results support the conclusions.
I have written some suggestions as a way to further improve the study. Below are my specific comments:
Line 2-4: Please make sure the title of the aricle is Palatino Linotype, 18, Bold
Line 67, 451, 454: C. albicans, Line 453, 469: A. terreus, Line 474, 507: A. nidulans etc. The names of the microorganisms must appear without abbreviation, when they appear for the first time in the text. Afterwards, it must appear only with abbreviations, which is not respected.
Line 140 - 141: Your research involves human material (urine sample and vaginal swab samples), There is a statement or agreement from the Research Ethics Committee of your institutions that the investigations were carried out in accordance with the rules of the Declaration of Helsinki of 1975, revised in 2013, although you said they are from the university collection..
Line151-152: Please specify what you used to measure the diameters in mm: ruler, electronic caliper, etc.
Line154 and 217: replace “thrice” with “three times”. Thrice it`s an “old -fashioned” way to say three times.
Line: 207-208: In the PCR part, you didn't say where you got the primer sequences from (a reference is needed).
Line 219: replace “half milliliter” with “0.5 mL”.
Line 219: Please use the word “hour” or check if “hr” is accepted as an abbreviation by the International System of Units (SI). I usually knew that “h” is the abbreviation.
Line 227: Agilent 1260, please add producer, city, country
Line 273-274: A. nidulans in italic
Line 277: C. albicans in italic
Line 283 and 288: In table 2, 3, 4 and 5 diameter 0 mm or ND – not detected, I think it is more appropriate to use the term R - resistant. R in the table and explain as a note under the table. The footer of the tables does not follow the format/instructions required by the journal.
Line 371: Delete one of the “in”
Line 405, 420: Please replace “&” with “and”
Line 407: Clear the gray color of the highlighted rows. The tables must comply with the format/instructions required by the journal.
Line 430, 442: S. aureus in italic
Line 433, 509: C. albicans in italic
Line 479, 506: Delete “total phenolic content” and replace with TPC, and where possible in the rest of the text.
Comments on the Quality of English Language
Only minor English editing is required.
Author Response
Response to Reviewer 1 Comments
Staphylococcus and Candida are acknowledged as etiological agents in a multitude of diseases, and the increasing prevalence of multidrug-resistant strains emphasizes the need to investigate natural sources, especially soil-fungi, as potential sources of antimicrobial agents. This study explores and demonstrates the remarkable efficacy of the antimicrobial potential of ethyl acetate extracts of soil-derived fungi against Staphylococcus and Candida. Noteworthy levels of efficiency were observed, especially in the case of the A. terreus AUMC 15447 and A. nidulans AUMC 15444 strains. Complementary analyzes and determinations support the authors' theory. The paper is interesting and multidisciplinary. The article was well documented and written, presenting a topic relevant to academia. I take this opportunity to congratulate the authors for their complex, excellent work. The article is interesting, however, there are small errors in the manuscript. However, in present form several explanations and modifications are needed. Overall experiments are well done, and the results support the conclusions.
I have written some suggestions as a way to further improve the study. Below are my specific comments:
Line 2-4: Please make sure the title of the aricle is Palatino Linotype, 18, Bold.
Response: Checked and corrected.
Line 67, 451, 454: C. albicans, Line 453, 469: A. terreus, Line 474, 507: A. nidulans etc. The names of the microorganisms must appear without abbreviation, when they appear for the first time in the text. Afterwards, it must appear only with abbreviations, which is not respected.
Response: Response: Checked and corrected throughout the whole manuscript.
Line 140 - 141: Your research involves human material (urine sample and vaginal swab samples), There is a statement or agreement from the Research Ethics Committee of your institutions that the investigations were carried out in accordance with the rules of the Declaration of Helsinki of 1975, revised in 2013, although you said they are from the university collection.
Response: Thanks for your valuable comment; these isolates were provided kindely by our colleagues and we maitain them with our laboratory culture collection. They already had a statement or agreement from The Ministry of Health Ethical Committee approved the study (ethical approval number 04), and we cite their article which included these strains.
Al Halteet, S.; Abdel-Hadi, A.; Hassan, M.; Awad, M. Prevalence and antifungal susceptibility profile of clinically relevant Candida species in postmenopausal women with diabetes. BioMed Res. Int. 2020, 2020, 7042490. https://doi.org/10.1155/2020/7042490
Line151-152: Please specify what you used to measure the diameters in mm: ruler, electronic caliper, etc.
Response: Specified (by Vernier caliper).
Line154 and 217: replace “thrice” with “three times”. Thrice it`s an “old -fashioned” way to say three times.
Response: Replaced.
Line: 207-208: In the PCR part, you didn't say where you got the primer sequences from (a reference is needed).
Response: We already state the reference “According to Mohamed et al. [72], internal transcribed spacer (ITS) region of the tested strain was amplified using specific universal primers ITS-1 (5′-TCC GTA GGT GAA CCT GCG G-3′) and ITS-4 (5′-TCC TCC GCT TAT TGA TAT GC-3′)”.
Mohamed, H.; El-Shanawany, A.; Shah, A.M.; Nazir, Y.; Naz, T.; Ullah, S.; Mustafa, K.; Song, Y. Comparative analysis of different isolated oleaginous Mucoromycota fungi for their γ-linolenic acid and carotenoid production. Biomed. Res. Int. 2020, 2020, 3621543. https://doi.org/10.1155/2020/3621543
Line 219: replace “half milliliter” with “0.5 mL”.
Response: Replaced.
Line 219: Please use the word “hour” or check if “hr” is accepted as an abbreviation by the International System of Units (SI). I usually knew that “h” is the abbreviation.
Response: Corrected.
Line 227: Agilent 1260, please add producer, city, country.
Response: Added “Agilent 1260 HPLC-DAD system (Agilent Technologies, Waldbronn, Germany) “.
Line 273-274: A. nidulans in italic
Response: Corrected.
Line 277: C. albicans in italic
Response: Corrected.
Line 283 and 288: In table 2, 3, 4 and 5 diameter 0 mm or ND – not detected, I think it is more appropriate to use the term R - resistant. R in the table and explain as a note under the table. The footer of the tables does not follow the format/instructions required by the journal.
Response: Thanks for your valuable comments and suggestions; they were considered and stated properly in the revised manuscript.
Line 371: Delete one of the “in”.
Response: Deleted.
Line 405, 420: Please replace “&” with “and”.
Response: Replaced.
Line 407: Clear the gray color of the highlighted rows. The tables must comply with the format/instructions required by the journal.
Response: Cleared.
Line 430, 442: S. aureus in italic.
Response: Corrected.
Line 433, 509: C. albicans in italic.
Response: Corrected.
Line 479, 506: Delete “total phenolic content” and replace with TPC, and where possible in the rest of the text.
Response: “Total phenolic content” was delete and replaced with TPC.

Reviewer 2 Report
Comments and Suggestions for Authors
We have thoroughly reviewed your manuscript and would like to offer some constructive feedback to enhance the clarity and depth of your research. Below are specific remarks for each section that we believe would improve the overall quality and impact of your article.
Introduction Section
-
- The introduction successfully outlines the general issue of drug resistance and the potential of fungal metabolites. However, it lacks a clear statement of the specific research gap your study aims to address. Clarifying this could strengthen the rationale for your research.
-
-
- We recommend expanding the introduction to explain how this study fits into the larger context of antimicrobial resistance and drug discovery. This would help in positioning your research within the broader field and underline its significance.
-
Paragraph Transition:
-
- The transition from paragraph 2 (line 64) to paragraph 3 needs to be smoother. Currently, it abruptly shifts from discussing secondary metabolites to systemic infections caused by Candida species. We suggest developing the paragraph further by focusing more on fungal metabolites, specifically those related to Candida, before transitioning. For instance: "The ability of Candida species to cause infections ranging from superficial to potentially life-threatening is well documented [27,28]. This variability in infection severity reflects the complexity and adaptability of these pathogenic organisms. Systemic infections caused by Candida spp. are particularly concerning due to their capacity to resist current antifungal treatments, thus underscoring the urgent need to develop new therapeutic strategies."
Materials and Methods
-
Sampling Protocol Clarity:
-
The sampling methodology should include specific details about the geographical locations and depths from which soil samples were collected.
-
Determination of Minimum Inhibitory Concentration:
-
Please provide details about the range of concentrations tested for each antimicrobial agent.
-
Statistical Analysis:
-
A more detailed description of the statistical methods and software used for data analysis is essential for replicability and validation of your results.
Results
-
3.1. Identification of Fungal Isolates:
-
Expanding on the specific criteria or features used for identifying fungal isolates based on macro- and microscopic characteristics would be helpful. Also, consider simplifying the sentence structure for better clarity, such as revising "Forty-five fungal isolates were obtained from different collected seven soil samples collected..." to a clearer form.
-
3.2. Antimicrobial Activities:
-
We suggest including a more detailed discussion about the variability in the efficacy of different extracts and their potential reasons. The data on MIC and MFC are crucial for understanding the clinical relevance of these extracts. A comparison with existing antimicrobial agents could further contextualize these findings. Additionally, a discussion on the potential mechanisms of action of these extracts, based on their chemical composition, would be insightful.
-
3.3. Antioxidant and Cytotoxic Activities:
-
The results are promising, but further studies, such as cell line-based cytotoxicity assays, are recommended to corroborate these findings and assess their clinical application potential.
General Remarks
-
Typographical Corrections:
- Line 255: Correct "anticandodal activities" to "antifungal activities" or "anti-Candida activities."
- Line 256: Correct "macro- and m microscopic" to "macro- and microscopic."
- Line 277: Write C. albicans in italics.
- Line 296: Write S. aureus in italics.
-
Consistency in Nomenclature: Ensure standard scientific nomenclature and maintain consistency throughout the article.
While the use of technical language is appropriate for the subject matter, ensure that it remains accessible to readers who may not be specialists in this specific field. Striking a balance between technical accuracy and readability is crucial.
Author Response
Response to Reviewer 2 Comments
We have thoroughly reviewed your manuscript and would like to offer some constructive feedback to enhance the clarity and depth of your research. Below are specific remarks for each section that we believe would improve the overall quality and impact of your article.
Introduction Section
- The introduction successfully outlines the general issue of drug resistance and the potential of fungal metabolites. However, it lacks a clear statement of the specific research gap your study aims to address. Clarifying this could strengthen the rationale for your research.
- We recommend expanding the introduction to explain how this study fits into the larger context of antimicrobial resistance and drug discovery. This would help in positioning your research within the broader field and underline its significance.
Response: Thanks for your valuable comment; these points was presented within the article context and supported as following with appropriate refrences in the revised manuscript.
“Growing fungal resistance limits their convenient therapeutic efficacies, thus driving the treatment of fungal infection disease more intractable [42]. MRSA strains are intrinsically resistant to β-lactams and rapidly developed resistance to multiple antimicrobial drug classes [43].”
Paragraph Transition
- The transition from paragraph 2 (line 64) to paragraph 3 needs to be smoother. Currently, it abruptly shifts from discussing secondary metabolites to systemic infections caused by Candida species. We suggest developing the paragraph further by focusing more on fungal metabolites, specifically those related to Candida, before transitioning. For instance: "The ability of Candida species to cause infections ranging from superficial to potentially life-threatening is well documented [27,28]. This variability in infection severity reflects the complexity and adaptability of these pathogenic organisms. Systemic infections caused by Candida spp. are particularly concerning due to their capacity to resist current antifungal treatments, thus underscoring the urgent need to develop new therapeutic strategies."
Response: Thanks for your valuable suggestion; it was considered and highlighted in the revised article.
Materials and Methods
Sampling Protocol Clarity
The sampling methodology should include specific details about the geographical locations and depths from which soil samples were collected.
Response: Thanks for your valuable comment; depths from which soil samples were collected “from the superficial layer of soil with the maximum depth of 10 cm”. The geographical locations was stated with (Latitude and longitude) “(27.8942°N, 42.6832°E)”.
Determination of Minimum Inhibitory Concentration
Please provide details about the range of concentrations tested for each antimicrobial agent.
Response: Two-fold serially diluted fungal extract was stated in this part illustrating the range of concentrations tested for each antimicrobial agent “The final fungal extract concentration, using two-fold serial dilution, in each well was 100, 50, 25, 12.5, 6.25, 3.125, 1.56, 0.78, 0.39, 0.195, and 0.098 mg/mL, respectively ”. The same concentrations were apllied for positive controls. It was considered and in the revised article.
Statistical Analysis
A more detailed description of the statistical methods and software used for data analysis is essential for replicability and validation of your results.
Response: A detailed description was done in the revised manuscript.
Results
3.1. Identification of Fungal Isolates:
Expanding on the specific criteria or features used for identifying fungal isolates based on macro- and microscopic characteristics would be helpful. Also, consider simplifying the sentence structure for better clarity, such as revising "Forty-five fungal isolates were obtained from different collected seven soil samples collected..." to a clearer form.
Response: Thanks for your valuable comments; the specific criteria or features used for identifying fungal isolates was explained and inserted in the materials and methods section “using the following key references, including Raper and Fennell [57], Moubasher [58] and Domsch et al. [59]”.
- “Forty-five fungal isolates were obtained from different collected seven soil samples collected... ” was revised to “Forty-five fungal isolates were derived from different collected soil samples ... ”.
3.2. Antimicrobial Activities:
We suggest including a more detailed discussion about the variability in the efficacy of different extracts and their potential reasons. The data on MIC and MFC are crucial for understanding the clinical relevance of these extracts. A comparison with existing antimicrobial agents could further contextualize these findings. Additionally, a discussion on the potential mechanisms of action of these extracts, based on their chemical composition, would be insightful.
Response: Thanks for your valuable comments; a proper discussion, regarding a comparison with existing antimicrobial agents, was stated and highlighted in the revised article .
- A convinent discussion on the potential mechanisms of action of phenolics and flavonids was presented in the last paragraph in the discussion section, where we deal with this study with detected phenolics and flavonids in different fungal extracts “Natural-derived substances are a prime contender for treating microbial infections…..”
3.3. Antioxidant and Cytotoxic Activities:
The results are promising, but further studies, such as cell line-based cytotoxicity assays, are recommended to corroborate these findings and assess their clinical application potential.
Response: Thanks for your valuable suggestion; further studies will be conducted including cell line-based cytotoxicity assays and in vivo experiments.
General Remarks
Typographical Corrections
- Line 255: Correct "anticandodal activities" to "antifungal activities" or "anti-Candida activities."
Response: Corrected.
- Line 256: Correct "macro- and m microscopic" to "macro- and microscopic."
Response: Corrected.
- Line 277: Write C. albicans in italics.
Response: Corrected.
- Line 296: Write S. aureus in italics.
Response: Corrected.
Consistency in Nomenclature:
Ensure standard scientific nomenclature and maintain consistency throughout the article.
Response: Ensured and and maintained throughout the article.

Reviewer 3 Report
Comments and Suggestions for Authors
Doctors Al Mousa and Hassane and their colleagues submitted a manuscript titled "Anti-Staphylococcal, Anti-Candida, Free Radical Scavenging Potential of Soil Fungal Metabolites: A Study Supported by Phenolic Characterization and Molecular Docking Analysis" for review.
These studies indicate a source of bioactive polyphenolic fungal metabolites useful in combating pathogenic staphylococci and yeasts, which may be useful in the development of a safe drug. The isolates of the soil microorganisms used were fully characterized in terms of the content of polyphenolic compounds, tests of their direct activity against pathogenic strains were carried out, and advanced tests were used, such as molecular docking of the targeted isolates to the key enzymes of pathogenic microorganisms, which is fully sufficient to publish the manuscript in the journal "CIMB", after removing a few typos, listed below:
At line 43 is: … thier … , but should be … their … .
At line 348 (Figure 1) is: ... A)Phylogenetic … , but should be … A) Phylogenetic
… . Comment: Please insert a space character after the bracket sign.
At line 399 is: … RMSD= 0.55 … , but should be better … RMSD = 0.55 … . Comment: Please insert a space before the equal sign. Please correct throughout the manuscript.
At line 498 is: … 3,4,5- methoxy cinnamic acid … , but most probably should be … 3,4,5-trimethoxycinnamic acid … . Comment: Please check and correct.
At line 501 is: … e-vanillic acid … . What is "e" vanillic acid? According to my knowledge, vanillic acid exists in one form - it cannot have geometric isomers or stereo-isomers.
The origin and purity of the reagents and solvents used in the tests are missing. Please add this information to enable the reader to reproduce the research results and facilitate the development of highly desirable drugs.
Comments on the Quality of English LanguageDoctors Al Mousa and Hassane and their colleagues submitted a manuscript titled "Anti-Staphylococcal, Anti-Candida, Free Radical Scavenging Potential of Soil Fungal Metabolites: A Study Supported by Phenolic Characterization and Molecular Docking Analysis" for review.
These studies indicate a source of bioactive polyphenolic fungal metabolites useful in combating pathogenic staphylococci and yeasts, which may be useful in the development of a safe drug. The isolates of the soil microorganisms used were fully characterized in terms of the content of polyphenolic compounds, tests of their direct activity against pathogenic strains were carried out, and advanced tests were used, such as molecular docking of the targeted isolates to the key enzymes of pathogenic microorganisms, which is fully sufficient to publish the manuscript in the journal "CIMB", after removing a few typos, listed below:
At line 43 is: … thier … , but should be … their … .
At line 348 (Figure 1) is: ... A)Phylogenetic … , but should be … A) Phylogenetic
… . Comment: Please insert a space character after the bracket sign.
At line 399 is: … RMSD= 0.55 … , but should be better … RMSD = 0.55 … . Comment: Please insert a space before the equal sign. Please correct throughout the manuscript.
At line 498 is: … 3,4,5- methoxy cinnamic acid … , but most probably should be … 3,4,5-trimethoxycinnamic acid … . Comment: Please check and correct.
At line 501 is: … e-vanillic acid … . What is "e" vanillic acid? According to my knowledge, vanillic acid exists in one form - it cannot have geometric isomers or stereo-isomers.
The origin and purity of the reagents and solvents used in the tests are missing. Please add this information to enable the reader to reproduce the research results and facilitate the development of highly desirable drugs.
Author Response
Response to Reviewer 3 Comments
Doctors Al Mousa and Hassane and their colleagues submitted a manuscript titled "Anti-Staphylococcal, Anti-Candida, Free Radical Scavenging Potential of Soil Fungal Metabolites: A Study Supported by Phenolic Characterization and Molecular Docking Analysis" for review.
These studies indicate a source of bioactive polyphenolic fungal metabolites useful in combating pathogenic staphylococci and yeasts, which may be useful in the development of a safe drug. The isolates of the soil microorganisms used were fully characterized in terms of the content of polyphenolic compounds, tests of their direct activity against pathogenic strains were carried out, and advanced tests were used, such as molecular docking of the targeted isolates to the key enzymes of pathogenic microorganisms, which is fully sufficient to publish the manuscript in the journal "CIMB", after removing a few typos, listed below:
At line 43 is: … thier … , but should be … their … .
Response: Thanks for your valuable comment; it was corrected.
At line 348 (Figure 1) is: ... A)Phylogenetic … , but should be … A) Phylogenetic
… . Comment: Please insert a space character after the bracket sign.
Response: Corrected.
At line 399 is: … RMSD= 0.55 … , but should be better … RMSD = 0.55 … .
Comment: Please insert a space before the equal sign. Please correct throughout the manuscript.
Response: Corrected.
At line 498 is: … 3,4,5- methoxy cinnamic acid … , but most probably should be … 3,4,5-trimethoxycinnamic acid … . Comment: Please check and correct.
Response: Corrected.
At line 501 is: … e-vanillic acid … . What is "e" vanillic acid? According to my knowledge, vanillic acid exists in one form - it cannot have geometric isomers or stereo-isomers.
Response: Sorry for this mistake, it was corrected.
The origin and purity of the reagents and solvents used in the tests are missing. Please add this information to enable the reader to reproduce the research results and facilitate the development of highly desirable drugs.
Response: Thanks for your valuable comment and suggestion; information about the used chemicals was provided within materials and methods section.

Reviewer 4 Report
Comments and Suggestions for Authors
Please see the attachment

Comments on the Quality of English LanguageMinor editing of English language required
Author Response
Response to Reviewer 4 Comments
General characterization of the manuscript
The reviewed manuscript is related to specific aspects of the general challenge of searching for fungal-derived antimicrobial compounds. Important virulence factors, as the secreted aspartic proteinase2 (SAP2) and tyrosyl-tRNA synthetase from C. albicans and S. aureus, respectively, are considered as the therapeutic targets to be implemented in the molecular docking analysis performed to examine phenolic phytochemicals as the compounds reducing the virulence severity.
There are several points to be addressed by the authors.
- There is one main problem related to the manuscript content: the poorly determined concentration of initial fungal extracts, on the basis of which the subsequent recalculations were done in the whole work for different purposes (section 2.3: the fermented medium was extracted twice by ethyl acetate, filtered over sodium sulphate anhydrous, and concentrated by a vacurotavapor).
The parameters of this initial extract should be bounded either with dry biomass (i.e. ml per dry weight), or a level of a (group) of compounds (may be of phenolics, or fatty acids, etc.).
Response: Thanks for your valuable comment; we already calculate the yield of of extracts for each fungal isolate as (g/ 100g) of the utilized rice medium (as a solid state fermentation), but we keep these results because we have to use it in further studies have been conducted on the basis of the results of the current article.
Other proposed corrections to manuscript
- In the Table 7, the expression of results as gallic acid equivalent for total phenolics, and quercetin equivalent for flavonoids, should be mentioned in the titles of columns.
Response: Gallic acid equivalent for total phenolics and quercetin equivalent for flavonoids, were mentioned in the titles of columns.
- In the Table 8, in contrast, the concentration expressed in the "μg/g" units is not clear, since the liquid solution injected in chromatograph was measured in microliters.
Response: Thanks for your valuable comment; we claculate the cconcentration ibased on the procedures for preparing extract by dissolving a known amount (mg) dry weight extract in the HPLC grade solvent (mL), so we can obtain the concentration of the detected compounds in regard to the initial quantity of the extract. However we clarified the expression in the table title.
